# Androglobin, a chimeric mammalian globin, is required for male fertility

Anna Keppner[1], Miguel Correia[1], Sara Santambrogio[2], Teng Wei Koay[1], Darko Maric[1], Carina Osterhof[3], Denise V Winter[4], Angèle Clerc[1], Michael Stumpe[5], Frédéric Chalmel[6], Sylvia Dewilde[7], Alex Odermatt[4], Dieter Kressler[5], Thomas Hankeln[3], Roland H Wenger[2], David Hoogewijs[1]*

[1]Department of Endocrinology, Metabolism and Cardiovascular system, University of Fribourg, Fribourg, Switzerland; [2]Institute of Physiology, University of Zurich, Zurich, Switzerland; [3]Institute for Organismic and Molecular Evolutionary Biology, University of Mainz, Mainz, Germany; [4]Department of Pharmaceutical Sciences, University of Basel, Basel, Switzerland; [5]Department of Biology, University of Fribourg, Fribourg, Switzerland; [6]University of Rennes, Inserm, UMR_S 1085, Rennes, France; [7]Department of Biomedical Sciences, University of Antwerp, Antwerp, Belgium

*For correspondence:
david.hoogewijs@unifr.ch

Competing interest: The authors declare that no competing interests exist.

**Abstract** Spermatogenesis is a highly specialized differentiation process driven by a dynamic gene expression program and ending with the production of mature spermatozoa. Whereas hundreds of genes are known to be essential for male germline proliferation and differentiation, the contribution of several genes remains uncharacterized. The predominant expression of the latest globin family member, androglobin (Adgb), in mammalian testis tissue prompted us to assess its physiological function in spermatogenesis. Adgb knockout mice display male infertility, reduced testis weight, impaired maturation of elongating spermatids, abnormal sperm shape, and ultrastructural defects in microtubule and mitochondrial organization. Epididymal sperm from Adgb knockout animals display multiple flagellar malformations including coiled, bifid or shortened flagella, and erratic acrosomal development. Following immunoprecipitation and mass spectrometry, we could identify septin 10 (Sept10) as interactor of Adgb. The Sept10-Adgb interaction was confirmed both *in vivo* using testis lysates and *in vitro* by reciprocal co-immunoprecipitation experiments. Furthermore, the absence of Adgb leads to mislocalization of Sept10 in sperm, indicating defective manchette and sperm annulus formation. Finally, *in vitro* data suggest that Adgb contributes to Sept10 proteolysis in a calmodulin-dependent manner. Collectively, our results provide evidence that Adgb is essential for murine spermatogenesis and further suggest that Adgb is required for sperm head shaping via the manchette and proper flagellum formation.

## Editor's evaluation

This manuscript demonstrates that male mice lacking androglobin, a poorly understood heme-containing protein, are infertile and have defects in late stage spermatogenesis. The revisions are thorough and the inclusion of additional data makes the manuscript solid. The lack of defects at the hypothalamus-pituitary level makes the phenotype more striking as a direct effect of the mutation at the testis level. Overall, this revised version is significantly improved.

## Introduction

Spermatogenesis is a complex and dynamic differentiation process ending by the production of mature haploid spermatozoa (*Hermo et al., 2010a*; *Hermo et al., 2010b*; *Hermo et al., 2010c*; *Hermo et al.,*

*2010d*; *Hermo et al., 2010e*). While type A spermatogonia undergo mitosis, one of the daughter cells serves to replenish the stem cell population, and the other daughter cell further divides mitotically, differentiates, and eventually forms type B spermatogonia. Following another mitotic division, type B spermatogonia engender primary spermatocytes, which complete their first meiotic division and form two secondary spermatocytes. Each secondary spermatocyte completes a second meiotic division, leading to the production of two haploid round spermatids (*Mecklenburg and Hermann, 2016*). During the final differentiation phase of spermatogenesis, known as spermiogenesis, the spermatids undergo profound morphological changes to differentiate and elongate into spermatozoa. These changes include condensation of the nucleus and compaction of the genetic material, the formation of the acrosome (the sperm head) from the Golgi apparatus before migration of the latter to the cytoplasmic droplet (*Khawar et al., 2019*), the formation of the sperm flagellum from the centriole, around which mitochondria will migrate to form the midpiece, the sperm annulus, and the mobile tail, and cytoplasmic reduction, whereby all unnecessary cytoplasmic remnants are eliminated. At the end of the elongation process, the spermatozoa are released into the lumen of the seminiferous tubule to migrate into the rete testis and epididymis for final maturation. Dysfunctions in any of these tightly orchestrated steps could lead to impaired spermatogenesis, meiotic arrest, or abnormal sperm formation with direct consequences on male fertility (*Neto et al., 2016*). It is currently estimated that sperm defects and abnormalities remain idiopathic in about 30% of cases (*Fainberg and Kashanian, 2019*; *Tüttelmann et al., 2018*), and unraveling their molecular basis appears challenging since it is believed that over 4000 genes are possibly implicated in spermatogenesis (*Jan et al., 2017*).

Globins are small globular metallo-proteins, which have the capacity to reversibly bind gaseous ligands via a typical 8 alpha-helical structure in which a heme prosthetic group can be embedded. In mammals, five globin types exist: the well-established hemoglobin and myoglobin, neuroglobin in neuronal cells, cytoglobin ubiquitously expressed in fibroblasts, and the more recently identified androglobin (Adgb), predominantly expressed in mammalian testis tissue (*Keppner et al., 2020*). Adgb is a chimeric protein containing an N-terminal calpain-like cysteine protease domain, followed by an uncharacterized 300 amino acid long region, a central permuted functional globin domain (*Bracke et al., 2018*), interrupted by a potential calmodulin (CaM)-binding IQ motif, and a large 700 amino acid long C-terminal tail of unknown identity (*Hoogewijs et al., 2012*). Intriguingly, the chimeric nature of this globin resembles the domain structure of globin-coupled sensors found in prokaryotes (*Hou et al., 2001*; *Thijs et al., 2007*). CaM is one of the most conserved proteins in eukaryotes and serves as a secondary messenger following intracellular binding of $Ca^{2+}$ and interaction with one of the more than 300 identified downstream target proteins (*Andrews et al., 2020*). CaM is the main intracellular receptor for $Ca^{2+}$, thereby participating in almost every biological process, including spermatogenesis and sperm maturation and function (*Darszon et al., 2011*). Decreased mRNA expression levels in infertile vs. fertile men (*Platts et al., 2007*) suggest a potential role of Adgb in spermatogenesis. Gene regulation and expression studies further suggest an association of Adgb with ciliogenesis including flagellum formation (*Koay et al., 2021*). However, the *in vivo* function of Adgb remains unexplored. In this study, we investigated the physiological function of Adgb during murine spermatogenesis by generating and analyzing Adgb knockout mice. We show that Adgb is mainly expressed in late steps of spermiogenesis, that it locates to the sperm flagellum, the annulus, and the midpiece, and that it is crucial for male fertility and sperm formation. Furthermore, we demonstrate that Adgb interacts with septin 10 (Sept10) and that co-localization is detectable within the sperm neck in stage 12 and stage 15 spermatids and within the annulus of stage 15 spermatids and mature sperm. Finally, *in vitro* data suggest that Adgb contributes to Sept10 proteolysis in a CaM-dependent manner.

## Results

### Adgb knockout mice display male infertility

A gene-trap strategy, provided by the Knockout Mouse Project (KOMP) (*Skarnes et al., 2011*), was applied to target exons 13 and 14 of the *Adgb* gene (*Figure 1—figure supplement 1A*). The correct targeting of ESCs was verified first by long-range PCR (*Figure 1—figure supplement 1B*) and second by Southern blotting (*Figure 1—figure supplement 1C*). The targeted *Adgb*^tm1a(KOMP)Wtsi allele (Adgb tm1a mice), generated on a C57BL/6N background, displays a gene-trap DNA cassette, which was inserted into the 12^th intron of the *Adgb* gene. The gene trap consists of a splice acceptor site,

an internal ribosome entry site, a β-galactosidase reporter sequence, and a neomycin resistance sequence. Breeding of Adgb tm1a mice with ubiquitously expressed CMV Cre-deleter mice allowed generation of mice deficient for exons 13 and 14 but still expressing the β-galactosidase reporter (Adgb tm1b mice) (*Figure 1—figure supplement 1A*). Furthermore, mating of Adgb tm1a mice with Flp-deleter mice enabled the generation of conditional floxed mice (Adgb tm1c) (*Figure 1—figure supplement 1A*). These mice were further crossed with CMV Cre-deleter mice to generate the full knockout animals (Adgb tm1d) (*Figure 1—figure supplement 1A*) that were used for all downstream applications if not otherwise stated. Genotyping was performed by regular PCR (*Figure 1—figure supplement 1D*) and revealed no significant differences in the Mendelian distribution of offspring for Adgb tm1d animals following interbreeding of heterozygous parents (380 pups: +/+, n=101; +/-, n=168; -/-, n=111; $X^2$=5.62, p>0.05, ns). The genetic ablation of Adgb expression was further verified by reverse transcription (RT)-quantitative (q)PCR (*Figure 1A*) and immunoblotting (*Figure 1B and C*). While female knockout mice displayed no fertility issues, male knockout mice never generated offspring, indicative for infertility. Full penetrance male infertility was also observed in homozygous tm1a and tm1b male mice, whereas homozygous tm1c animals showed normal fertility (data not shown). Accordingly, the testis weight was significantly reduced in knockouts (*Figure 1D*). Intratesticular testosterone (*Figure 1E*) as well as serum luteinizing hormone (LH) and follicle-stimulating hormone (FSH) levels (*Figure 1—figure supplement 2*) remained comparable between wild type and knockouts. Stage-specific histological examination of seminiferous tubules of control animals revealed normal architecture, normal spermatogenic maturation steps, and the presence of mature sperm with flagella extending into the lumen (*Figure 1F*). In contrast, in knockout animals, despite the presence of meiotic events (stages X–XII, *Figure 1F*), no flagella could be observed during the spermiation stage (stages VII–VIII, *Figure 1F*). The absence of mature sperm was accompanied by abnormally shaped heads, trapped stage 16 spermatids within the epithelium, and the presence of cytoplasmic material filling the lumen of the tubules (*Figure 1F*). Within the cauda epididymis, knockout animals displayed accumulations of residual bodies, cytoplasmic material, shed germ cells, and occasional abnormally shaped sperm heads but an overall absence of mature sperm as compared to wild-type animals (*Figure 1G*). No differences were detected in knockout testes at mRNA levels of nitric oxide synthases 1–3 (*Nos1-3*) or superoxide dismutases 1–3 (*Sod1-3*), the ratio of *Bax/Bcl2* was unchanged compared to wild-type testes, and terminal deoxynucleotidyl transferase dUTP nick-end labeling (TUNEL) assays on testis sections revealed no differences, suggesting no increase in oxidative stress or apoptotic events (*Figure 1—figure supplement 3*).

## Absence of Adgb interferes with the maturation of elongating spermatids

To determine the temporal expression pattern of *Adgb* during spermatogenesis, wild-type embryos and pups at different post-natal ages, corresponding to the stages of the first wave of spermatogenesis during puberty, were dissected and analyzed by RT-qPCR. Whereas the expression of *Adgb* remained nearly undetectable until post-natal day 21 (corresponding to the stage of round spermatids), *Adgb* mRNA levels drastically increased to reach a peak at post-natal day 25, coinciding with the first elongating spermatids (*Figure 2A*). Bulk and single-cell RNA sequencing (scRNAseq) analysis in mouse and human datasets available at the ReproGenomics Viewer resource (*Darde et al., 2019*; *Darde et al., 2015*) as well as more recent murine scRNAseq data (*Kwak and Jung, 2019*) confirmed the conserved expression pattern of *ADGB* in spermatids (*Green et al., 2018*; *Jégou et al., 2017*; *Lukassen et al., 2018*; *Wang et al., 2018*; *Figure 2—figure supplement 1*; *Figure 2—figure supplement 2*). Accordingly, Adgb protein expression equally reached its peak at post-natal days 26–28, suggesting slightly delayed translation (*Figure 2B and C*). This finding was further confirmed by propidium iodide staining and FACS sorting on testis lysates of the different genotypes. While no variations could be detected for phases 2C (spermatogonia, secondary spermatocytes, and testicular somatic cells), S-phase (pre-meiotic spermatogonia), 4C (primary spermatocytes), and 1C (round spermatids), an abnormal accumulation of elongating and elongated spermatids (phase H) could be detected in knockout animals, suggesting a blockade in the elongation process (*Figure 2D*). Additionally, immunofluorescence (*Figure 2E*), mRNA *in situ* hybridization (*Figure 2F*), and X-gal (*Figure 2G*) stainings of testis sections from wild-type and knockout mice confirmed the presence of Adgb within layers containing post-meiotic cells and further intensifying toward the lumen and mature sperm

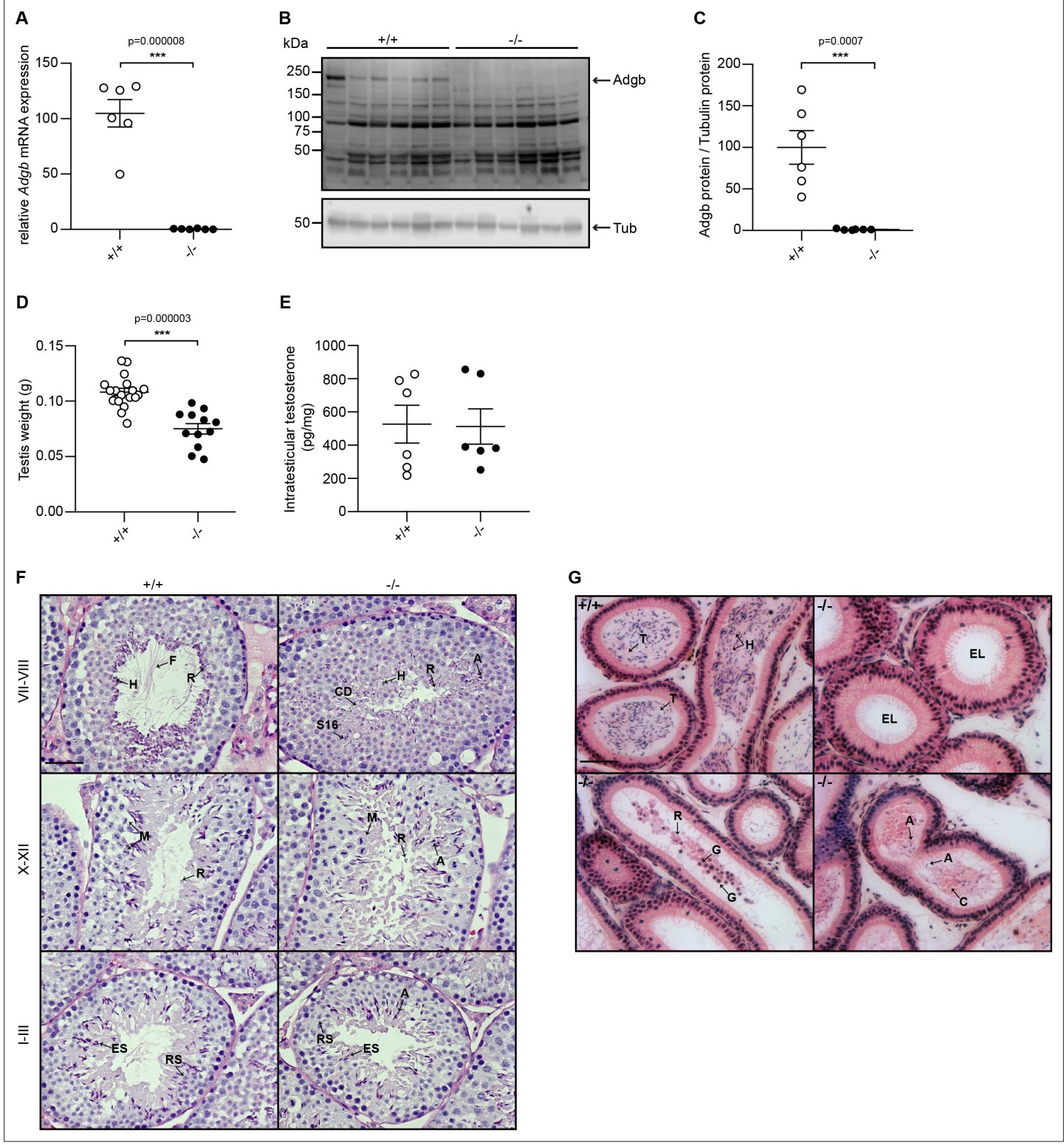

**Figure 1.** Validation of the knockout model and testicular phenotype. (**A**) Relative mRNA expression levels of *Adgb* in testes of wild-type (+/+) and knockout mice (-/-) (n=6 per genotype; p=0.000008). (**B**) Representative immunoblot for Adgb in testis lysates from wild-type (+/+) and knockout mice (-/-) (n=6 per genotype) and (**C**) corresponding protein quantification. Tubulin was used as loading control. p=0.0007. (**D**) Testis weight (g) in Adgb wild-type (+/+), heterozygous (+/-), and knockout (-/-) mice (n=8–13 per genotype). p=0.000003. (**E**) Intratesticular testosterone levels (pg/mg) in Adgb wild-type (+/+) and knockout (-/-) mice (n=6 per genotype). (**F**) Representative periodic acid Schiff (PAS)-hematoxylin-stained sections of testes from Adgb wild-type (+/+) and knockout mice (-/-) at the different stages of spermatogenesis. Heads (H), flagella (F), residual bodies (R), cytoplasmic debris

*Figure 1 continued on next page*

*Figure 1 continued*

(CD), meiosis (M), elongating spermatids (ES), round spermatids (RS), stage 16 spermatids (S16), and abnormal heads (A) are indicated. Note the full absence of flagella in knockout sections. Scale bar represents 50 µm. (**G**) Representative H&E stained sections of epididymides from Adgb wild-type (+/+) and knockout mice (-/-). H, tails (T), cytoplasmic bodies (C), R, germ cells (G), and A are shown. Note the empty lumen (EL) in knockout mice. Scale bar represents 50 µm. ** p<0.01, *** p<0.001.

The online version of this article includes the following source data and figure supplement(s) for figure 1:

**Source data 1.** Original uncropped immunoblots of *Figure 1B* with indication of the cropped areas.

**Figure supplement 1.** Generation of Adgb knockout mice.

**Figure supplement 1—source data 1.** Original uncropped gels of *Figure 1—figure supplement 1B,C,D* with indication of the cropped areas.

**Figure supplement 2.** Normal serum gonadotropin levels in Adgb knockout mice.

**Figure supplement 3.** Adgb knockout mice do not display changes in *Nos, Sod,* and apoptotic gene expression.

---

(*Figure 2E–G*). Moreover, in mature sperm, Adgb expression could be visualized within the midpiece and along the whole flagellum by both X-gal (*Figure 2G*) and immunofluorescence (*Figure 2H*) stainings.

## Adgb is required for proper sperm flagellum formation

To gain additional insights into the origin of male infertility, cauda epididymal sperm was collected from both wild-type and knockout mice, and visualized under a microscope. While wild-type sperm appeared normal, very few knockout spermatozoa were found and displayed various defects of the head and/or flagellum structure, including shortened or bifid tails, loopings of the flagellum, and immature acrosomal structures (*Figure 3A*). Knockout sperm acrosome structure appeared partially conserved, with sperm displaying either normal acrosomes, or various defects in shape or staining intensity, or total absence (*Figure 3B*). Stage-specific transmission electron microscope (TEM) ultrastructural analysis of testis sections revealed various defects associated with sperm flagellum formation. Whereas numerous axonemes displaying the regular 9+2 structure from step 9 spermatids could be found in wild types, none were found in knockouts at the same stage. In later stages (step 12 spermatids and onward), rare and abnormal axonemes could be observed in knockouts, displaying disorganized microtubular structures and forming microtubular clusters without defined fibrous or mitochondrial sheath (*Figure 3C*). We could further observe misshaped heads with nuclear inclusions, defective manchette elongation, and abnormal acrosomes in knockout sections (*Figure 3C–F*).

## The Adgb-dependent transcriptome reveals dysregulation of multiple spermiogenesis genes

To understand the molecular consequences of loss of Adgb in the testis, we performed RNAseq experiments on total testis RNA from wild-type and knockout mice at post-natal day 25. An elaborate set of significantly differentially expressed genes (74 genes upregulated and 204 downregulated) was identified, underscoring the crucial function of Adgb in spermatogenesis (*Figure 3—figure supplement 1A*, *Figure 3—source data 1*). Functional analysis based on gene ontology term enrichments confirmed that many of these genes are related to sperm head, acrosome reaction, acrosomal membrane, sperm motility, spermatid development, and spermatid differentiation, in line with the pronounced structural changes in spermatids during spermiogenesis (*Figure 3—figure supplement 1B*).

## Adgb interacts and co-localizes with Sept10

To obtain more insights into the physiological function of Adgb, we explored the Adgb-dependent interactome. Total protein extracts from wild-type testes were immunoprecipitated (IP) with anti-Adgb or control IgG antibodies and subsequently submitted to mass spectrometry (MS) analysis to reveal potential interacting proteins of Adgb. Among the specifically enriched proteins, there were various members of the septin family, such as Sept10, Sept11, Sept2, and Sept7 (*Figure 4A* and *Figure 4—source data 4*). Particular focus was put on Sept10 for further downstream experiments given its strong enrichment (4.509 log2-fold) combined with high abundance (3.864 log2[mean]) in the immunoprecipitation as well as substantial sequence coverage (40.5%), all reflected by its close position to Adgb among all septins in *Figure 4A*. To confirm the interaction between Adgb and

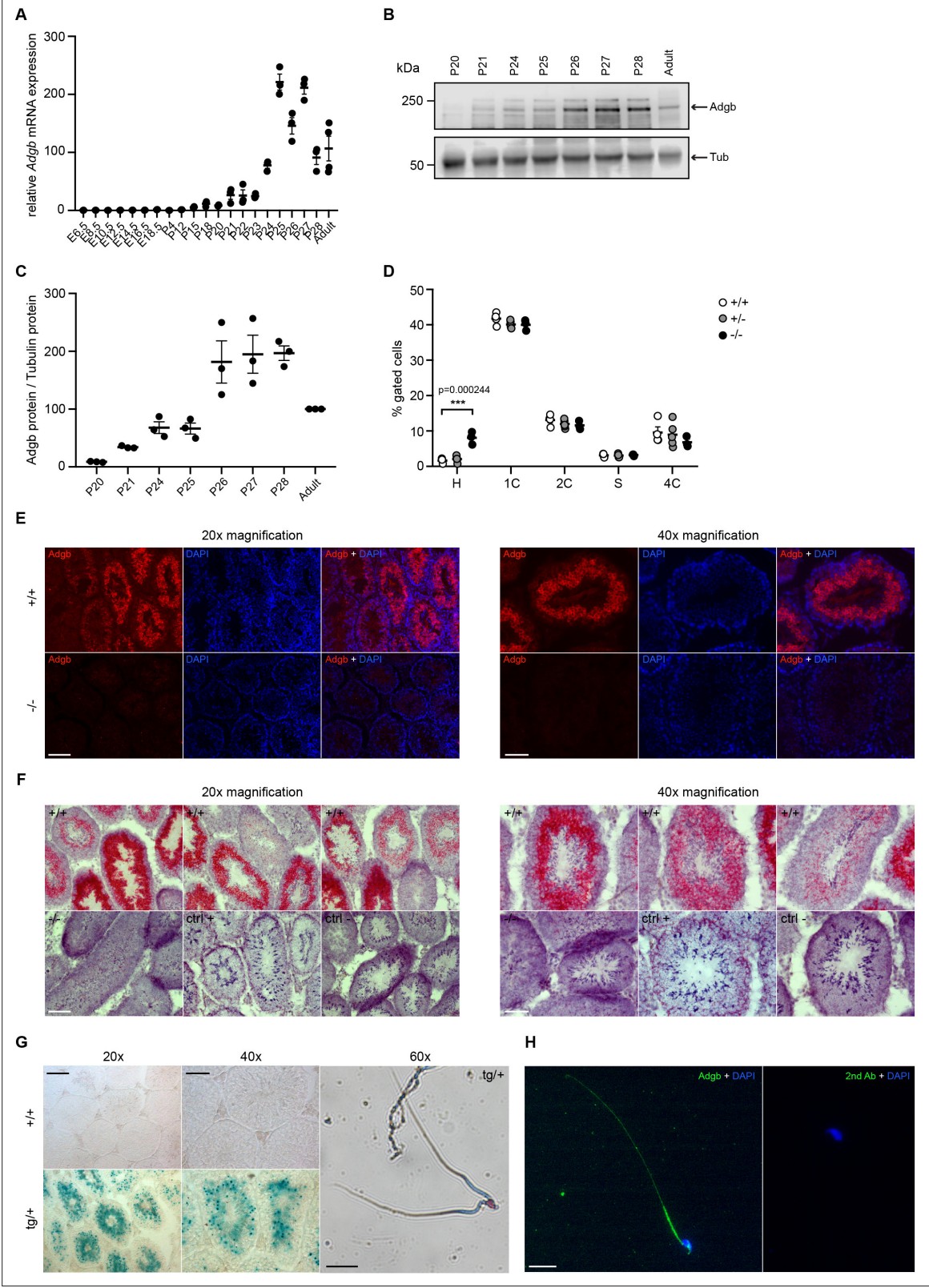

**Figure 2.** Testicular Adgb expression pattern and localization. (**A**) Relative mRNA expression levels of *Adgb* in testes of wild-type mice during embryonic development (E) and early post-natal (P) life (n=3–4 per condition). (**B**) Representative immunoblot for Adgb in testis lysates from wild-type mice at different P ages (n=3 per condition) and (**C**) corresponding protein quantification. Tubulin was used as loading control. (**D**) Flow cytometric analysis of spermatogenic cell populations following propidium iodide staining in Adgb wild-type (+/+, white circles, n=4), heterozygous (+/-, gray

*Figure 2 continued on next page*

*Figure 2 continued*

circles, n=5), and knockout (-/-, black circles, n=3) testes. H: elongating and elongated spermatids; 1 C: round spermatids; 2 C: spermatogonia, secondary spermatocytes, and testicular somatic cells; S: spermatogonia synthesizing DNA; 4 C: primary spermatocytes. p=0.00024. (**E**) Representative pictures of Adgb protein (red fluorescence) detection in testes of wild-type (+/+) and knockout (-/-) animals. Left panels 20× magnification, right panels 40× magnification, and scale bars represent 100 µm and 50 µm, respectively. Nuclei were stained with DAPI. (**F**) Representative pictures of *Adgb* mRNA *in situ* hybridization in testes from wild-type (+/+) and knockout (-/-) animals. Left panels 20× magnification; right panels, 40× magnification; scale bars represent 100 µm and 50 µm, respectively. Positive (ctrl +, PPIB) and negative (ctrl −, DapB) control sections are shown. (**G**) Representative pictures of β-galactosidase activity (X-gal staining) in testes from Tm1b wild-type (+/+) and Tm1b heterozygous (tg/+) mice and isolated spermatozoa from Tm1b heterozygous (tg/+) mice. Left panels, 20× magnification; middle panels, 40× magnification; right panel, 60× magnification; scale bars represent 100 µm, 50 µm, and 20 µm, respectively. Spermatozoa were counterstained with nuclear fast red. (**H**) Representative picture of Adgb protein (green fluorescence) in a single spermatozoon from wild-type (+/+) mice (left panel) and negative control (secondary antibody only, right panel). Scale bar represents 20 µm and nuclei were stained with DAPI. ** p<0.001.

The online version of this article includes the following source data and figure supplement(s) for figure 2:

**Source data 1.** Original uncropped immunoblots of *Figure 2B* with indication of the cropped areas.

**Figure supplement 1.** Temporal *Adgb* and *Sept10* expression profiles based on single-cell and bulk RNAseq datasets.

**Figure supplement 2.** *Adgb* and *Sept10* gene expression profiles based on single-cell RNAseq of mouse testes along the temporospatial axis of spermatid maturation.

Sept10, reciprocal co-immunoprecipitation (co-IP) experiments were performed (*Figure 4B and C*) on tissue extracts from wild-type and knockout testes (*Figure 4B*) and in HEK293 cells overexpressing full-length ADGB (*Figure 4C*) and SEPT10. The results demonstrate that Adgb and Sept10 interact both *in vivo* (*Figure 4B*) and *in vitro* (*Figure 4C*), whereas in testis lysates of Adgb-deficient mice, no Sept10 co-precipitation was observed (*Figure 4B*). Endogenous Sept10 protein levels were equal in testis lysates of Adgb-deficient and wild-type mice, as were endogenous levels of Sept11, Sept7, and Sept2, as well as other septins that are crucial for spermatogenesis, including Sept8, Sept9, and Sept14 (*Figure 4—figure supplement 1*). Reciprocal co-IP experiments between Adgb and other enriched septins (Sept2, Sept7, and Sept11) revealed no interaction, neither *in vivo* nor *in vitro* (*Figure 4—figure supplement 2*, *Figure 4—figure supplement 3*), whereas SEPT7 and SEPT10 interacted *in vitro* (*Figure 4—figure supplement 3*). We next investigated whether the interaction with SEPT10 occurs at the N-terminal or the C-terminal portion of ADGB (*Figure 4D and E*). Following co-overexpression of ADGB deletion constructs (*Figure 4—figure supplement 4*) with SEPT10 and subsequent co-IP, immunoblotting revealed that both parts of ADGB interact with SEPT10 (*Figure 4D and E*) and that this interaction remained intact also upon deletion of the coiled-coil domain of ADGB (*Figure 4—figure supplement 5*).

Consistent with a functional interaction, the temporal expression profiles of *SEPT10* and *ADGB* substantially overlap as illustrated by analysis of bulk and scRNAseq datasets of mouse and human RNA (*Figure 2—figure supplement 1*; *Figure 2—figure supplement 2*) as well as by RT-qPCR and immunoblotting of mouse tissue samples (*Figure 4—figure supplement 6*). The localization of Adgb and Sept10 was assessed in intact testis sections, microdissected tubules, and in epididymal sperm by immunofluorescence. Co-localization of Adgb and Sept10 was visible during spermatid maturation and in flagella in testis sections (*Figure 5A*) and at the level of the sperm annulus in S12 and S15 spermatids and mature wild-type sperm (*Figure 5B*, *Figure 5—figure supplement 1A*). A moderate Sept10 staining was also observed in the neck region of S15 spermatids and in mature sperm (*Figure 5B*, *Figure 5—figure supplement 1A*). The localization of Sept10 in knockout testis sections appeared overall fainter but within the same locations as in wild types, with the exception of sperm flagella (*Figure 5A*). However, in knockout S12 and S15 spermatids, as well as in epididymal sperm, only a single signal, likely corresponding to the annulus as verified by Sept7 staining of epididymal sperm (*Figure 5—figure supplement 1A*), was observed and displayed abnormal migration, indicating defective manchette or microtubule formation (*Figure 5B*, *Figure 5—figure supplement 1A*). The migration of the annulus drives mitochondrial placement along the forming mid-piece (*Toure et al., 2011*). Since knockout animals displayed abnormal ultrastructural mitochondria organization (*Figure 3*), CoxIV staining was performed on wild-type and knockout microdissected tubules and epididymal sperm. As expected, a robust staining was observed along the whole midpiece in wild-type spermatids and sperm, whereas mitochondria were barely visible and formed a cloudy structure around the neck region of knockout spermatids and sperm (*Figure 5—figure supplement 1B*).

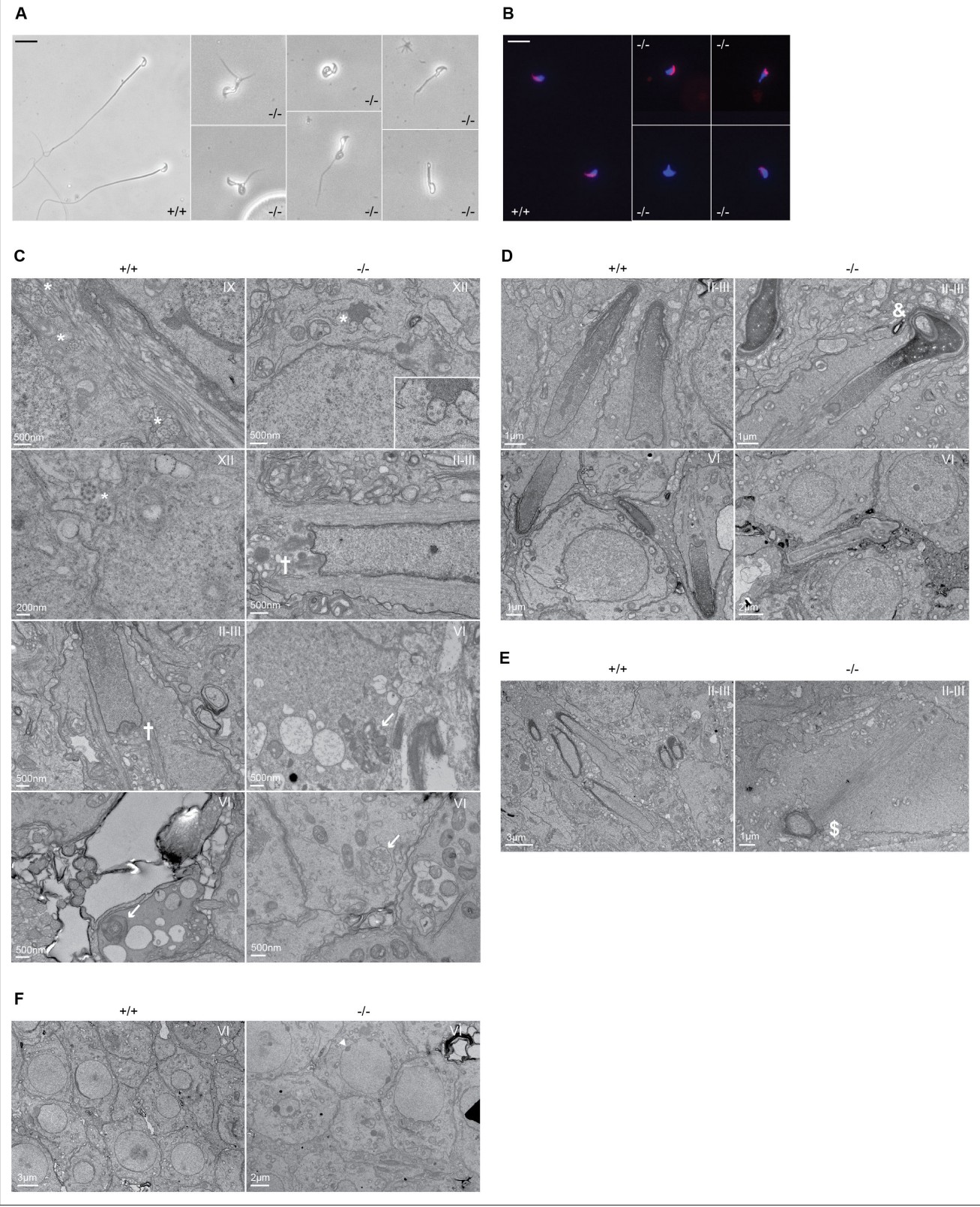

**Figure 3.** Defective spermatogenesis is associated with flagellar malformation in Adgb knockout mice. (**A**) Representative pictures of cauda epididymis sperm from wild-type (+/+) and Adgb knockout animals (-/-). Scale bar represents 20 μm. (**B**) Representative pictures of peanut agglutinin-stained cauda epididymis sperm from wild-type (+/+) and Adgb knockout animals (-/-). Nuclei were stained with DAPI. Scale bar represents 20 μm. (**C–F**) Representative transmission electron microscope (TEM) pictures from wild-type (+/+, left panels) and knockout (-/-, right panels) testes at various

*Figure 3 continued on next page*

*Figure 3 continued*

stages of the first wave, tubular stages are indicated. (**C**) 9+2 microtubular structure (asterisks), forming sperm flagella (crosses), and (impaired) outer dense fibers (arrows) are shown. (**D**) Misshaped sperm heads with nuclear inclusions (ampersand), (**E**) defective manchette elongation (dollar), and (**F**) abnormal acrosomes (arrowheads) are shown. Scale bar lengths are indicated on each picture.

The online version of this article includes the following source data and figure supplement(s) for figure 3:

**Source data 1.** Differentially regulated genes in wild-type vs. Adgb knockout testes.

**Figure supplement 1.** Volcano plot and gene ontology (GO) term analysis of differentially expressed genes in Adgb knockout mice testis samples.

## ADGB contributes to SEPT10 proteolytic cleavage *in vitro*

To analyze the functional consequences of the SEPT10-ADGB interaction, we transiently co-over-expressed both proteins in HEK293 cells. Intriguingly, apart from the intact form of overexpressed SEPT10 at 60 kDa, increased levels of a lower band of 37 kDa were consistently detected in the presence of co-overexpressed ADGB in a dose-dependent manner (*Figure 6A*). Immunoblotting with a V5-antibody upon co-overexpression of a C-terminal V5-tagged SEPT10 with ADGB (*Figure 6B*) as well as the presence of this band upon SEPT10/ADGB co-IP (*Figure 6B*, *Figure 6—figure supplement 1*) further supports its origin as proteolytic SEPT10 product. To investigate a potential oxygen-dependent influence of the globin domain, this experiment was repeated under normoxic and hypoxic conditions (0.2% $O_2$) but no differences were observed upon exposure to hypoxic conditions (*Figure 6C*). To determine a potential role of CaM, we constructed a deletion mutant lacking the IQ domain. Notably, transient overexpression of IQ-mutant ADGB resulted in considerably reduced appearance of the 37 kDa SEPT10 band relative to wild-type ADGB (*Figure 6D*), and the same was observed with an ADGB protease domain deletion mutant, further supporting a CaM-dependent proteolytic cleavage (*Figure 6D*). These findings prompted us to experimentally validate the suspected CaM-ADGB interaction. Whereas co-IP experiments following overexpression of full-length ADGB did not interact with CaM under basal experimental conditions in HEK293 cells (*Figure 6—figure supplement 2A*), a truncated construct covering the globin and IQ domains displayed robust ADGB-CaM interaction (*Figure 6E*). Consistently, MS analysis of proteins that were present in the IP of the overexpressed, isolated globin domain revealed a prominent enrichment of endogenous CaM (*Figure 6—figure supplement 3* and *Figure 6—source data 5*). Importantly, individual or double mutation of the proximal histidine (HisF8) or distal glutamine (GlnE7), critical residues in the globin domain for heme coordination, did not alter the interaction, suggesting that the ADGB-CaM interaction occurs independently of heme incorporation (*Figure 6F*, *Figure 6—figure supplement 4*). To fully exclude that the mutant globin domain might still be hemylated, we repeated these experiments under medium heme depletion and heme synthesis blocking conditions. Consistently, upon heme deprivation, robust interaction was observed between CaM and the isolated intact as well as double mutated (HisF8/GlnE7) globin domain. (*Figure 6—figure supplement 5*). As an additional layer of support for the ADGB-CaM interaction, chimeric Gal4 DNA-binding domain and VP16 transactivation domain-based fusion constructs were generated for mammalian 2-hybrid (M2H) luciferase reporter gene assays (*Figure 6G and H*). These M2H assays were performed in two different cell lines, HEK293 and A375, and revealed up to 7.5-fold increase in luciferase activity upon co-transfection of both chimeric proteins compared to single construct transfections, providing independent evidence that ADGB interacts with CaM. However, when full-length ADGB was employed in M2H assays with interchanged Gal4 and VP16 domains, no interaction with CaM was observed, corroborating the co-IP data (*Figure 6—figure supplement 2B*). To further assess the potential $O_2$-dependency of the CaM-ADGB interaction, we repeated the M2H assays in A375 cells under hypoxic conditions. Exposure to hypoxic conditions (0.2% $O_2$) did not alter luciferase activity (*Figure 6H*, left panel), while luciferase activity of a 5'/3'-hypoxia response element-dependent *EPO* promoter-driven reporter gene increased (*Figure 6H*, right panel), suggesting that the ADGB-CaM interaction is $O_2$ independent. Of note, while maintaining the $O_2$-independent interaction between Gal4-CaM and VP16-globin, the single Gal4-CaM construct control was increased under hypoxic conditions, albeit to a much lesser extent than the *EPO* control, possibly related to increases of cytoplasmic $Ca^{2+}$ mobilized from intracellular stores or by extracellular influx under hypoxic conditions and according activation of CaM (*Yuan et al., 2005*). This result is consistent with the maintained ADGB-CaM interaction following mutation of critical residues in the globin domain in co-IP experiments and unchanged ADGB-dependent

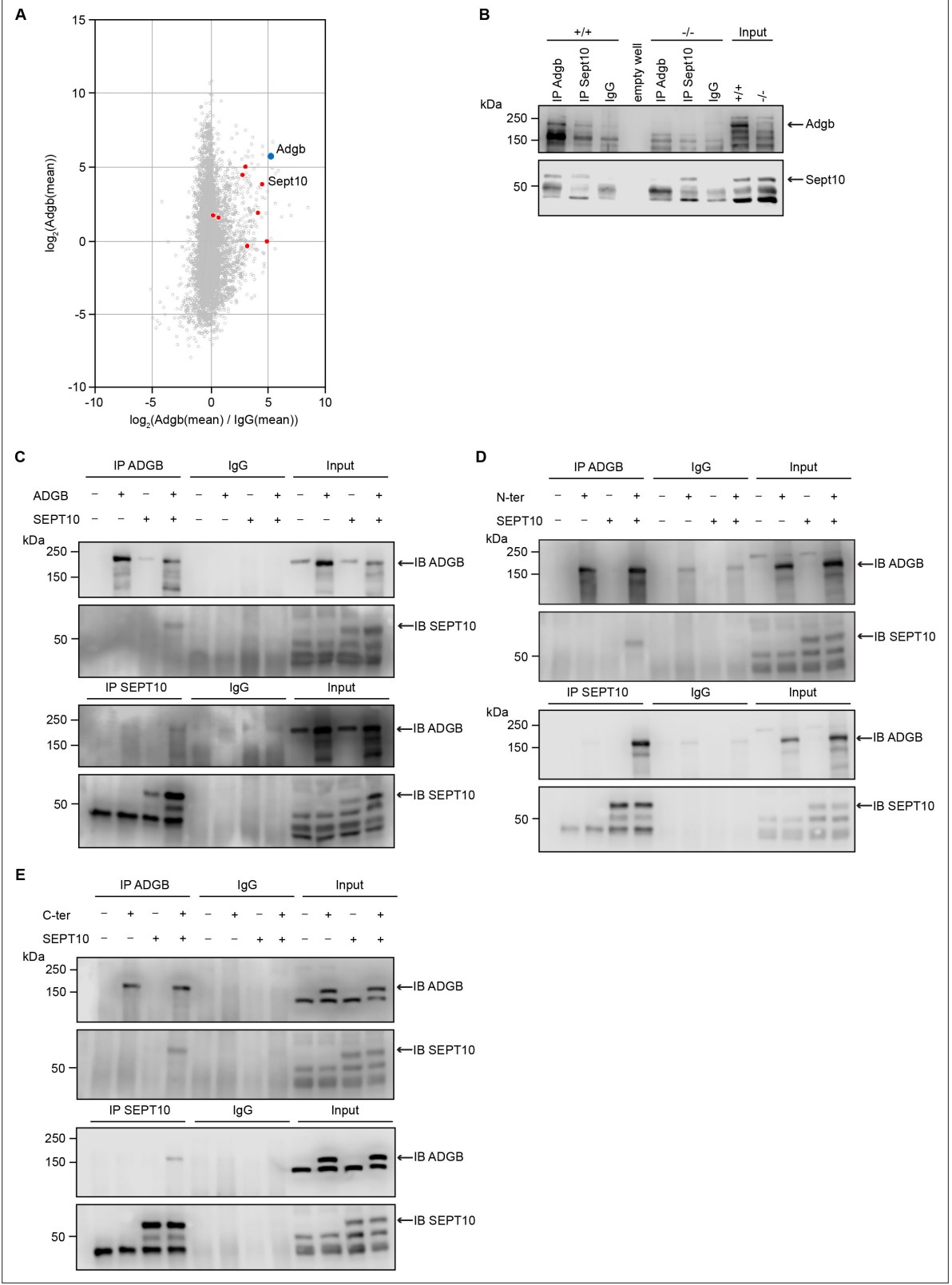

**Figure 4.** Adgb and Sept10 interact *in vivo* and *in vitro*. (**A**) Proteins of the septin family are specifically enriched in the Adgb immunoprecipitation (IP). The iBAQ (intensity-based absolute quantification) values of each Adgb IP (triplicate) and IgG control IP (duplicate) were log2 transformed and normalized against the median value. Missing values were imputed before the mean values of the Adgb and IgG control IPs were calculated. The normalized abundance of each protein detected in the Adgb IP (log2 Adgb [mean]) is plotted against its specific enrichment compared to the IgG

*Figure 4 continued on next page*

*Figure 4 continued*

control IP log2 (Adgb [mean]/IgG [mean]). Adgb and septins are highlighted as blue and red dots, respectively, in the christmas tree plot representation. (**B**) Representative immunoblot of Adgb and Sept10 in testis lysates from wild-type (+/+) and knockout (-/-) mice following co-IP of Adgb and Sept10. (**C–E**) Representative immunoblots of ADGB and SEPT10 in protein lysates of HEK293 cells (co-)transfected with full-length ADGB (C), N-ter ADGB (D) and C-ter ADGB (E), and SEPT10 following co-IP of ADGB and SEPT10. Schematic representation of deletion constructs is provided in *Figure 4—figure supplement 4*.

The online version of this article includes the following source data and figure supplement(s) for figure 4:

**Source data 1.** Original uncropped immunoblots of *Figure 4B and C* with indication of the cropped areas.

**Source data 2.** Original uncropped immunoblots of *Figure 4D* with indication of the cropped areas.

**Source data 3.** Original uncropped immunoblots of *Figure 4E* with indication of the cropped areas.

**Source data 4.** Raw mass spectrometry (MS) data of the Adgb immunoprecipitation (IP) vs. IgG control IP.

**Figure supplement 1.** The protein expression levels of Sept2, Sept7, Sept8, Sept9, Sept10, Sept11, and Sept14 are unaffected in Adgb knockout testis.

**Figure supplement 1—source data 1.** Original uncropped immunoblots of *Figure 4—figure supplement 1A-C* with indication of the cropped areas.

**Figure supplement 1—source data 2.** Original uncropped immunoblots of *Figure 4—figure supplement 1D-F* with indication of the cropped areas.

**Figure supplement 1—source data 3.** Original uncropped immunoblots of *Figure 4—figure supplement 1G* with indication of the cropped areas.

**Figure supplement 2.** Adgb does not interact with other septin family members *in vivo*.

**Figure supplement 2—source data 1.** Original uncropped immunoblots of *Figure 4—figure supplement 2A,B,C* with indication of the cropped areas.

**Figure supplement 3.** ADGB does not interact with other septin family members *in vitro*.

**Figure supplement 3—source data 1.** Original uncropped immunoblots of *Figure 4—figure supplement 3A* with indication of the cropped areas.

**Figure supplement 3—source data 2.** Original uncropped immunoblots of *Figure 4—figure supplement 3B* with indication of the cropped areas.

**Figure supplement 3—source data 3.** Original uncropped immunoblots of *Figure 4—figure supplement 3C* with indication of the cropped areas.

**Figure supplement 3—source data 4.** Original uncropped immunoblots of *Figure 4—figure supplement 3D* with indication of the cropped areas.

**Figure supplement 3—source data 5.** Original uncropped immunoblots of *Figure 4—figure supplement 3E* with indication of the cropped areas.

**Figure supplement 4.** ADGB constructs used throughout the study.

**Figure supplement 5.** The interaction between ADGB and SEPT10 is maintained despite mutation of the coiled-coil domains.

**Figure supplement 5—source data 1.** Original uncropped immunoblots of *Figure 4—figure supplement 5* with indication of the cropped areas.

**Figure supplement 6.** Sept10 temporal expression profile on mRNA and protein levels.

**Figure supplement 6—source data 1.** Original uncropped immunoblots of *Figure 4—figure supplement 6* with indication of the cropped areas.

cleavage of SEPT10 under hypoxic conditions. Collectively, these *in vitro* data suggest a scenario in which overexpressed ADGB proteolytically contributes to cleavage of overexpressed SEPT10 in an $O_2$-independent but CaM-dependent manner.

## Discussion

In the present study, we explored the testicular function of Adgb by generating and analyzing Adgb constitutive knockout mice. Our results demonstrate that Adgb is indispensable for proper sperm formation and male fertility. The absence of Adgb has profound deleterious effects on the elongation of spermatids, leading to multiple malformations of the sperm flagellum and head, as evidenced by the presence of various ultrastructural defects, including aberrant microtubule formation, misshaped heads with nuclear inclusions, defective manchette formation, and abnormal acrosomes. Notably, Adgb could be localized not only to the midpiece and sperm flagellum but also in the neck region (likely the centriole) and within the sperm annulus.

In principle, the observed male infertility could be the result of defects in the hypothalamic-pituitary-gonadal (HPG) axis. Unchanged LSH and LH levels indicate an intact HPG signaling and suggest the phenotypes represent direct (primary) effects of Adgb deficiency in testis. However, indirect effects cannot be entirely excluded despite normal serum levels of gonadotropins. Unchanged testicular testosterone levels are consistent with the total absence of Adgb expression in Leydig cells of wild-type mice.

Strikingly, upon LC-MS/MS interactome analysis after Adgb immunoprecipitation in testicular lysates, several members of the septin family of proteins ranked among the top hits. Septins are

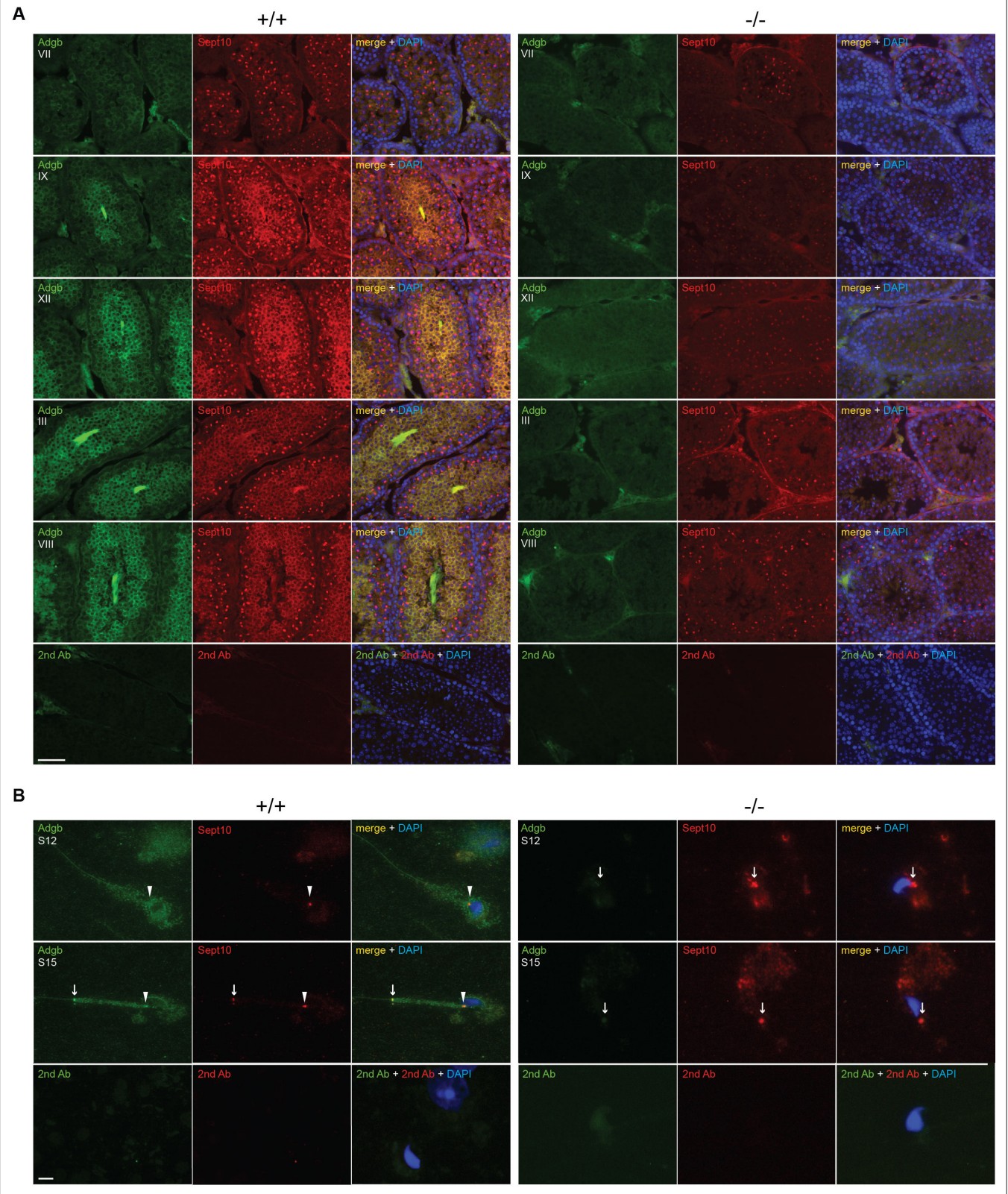

**Figure 5.** Adgb and Sept10 co-localize in the sperm neck and annulus. (**A**) Representative pictures of Adgb protein (green fluorescence) and Sept10 (red fluorescence) in testis sections from wild-type (+/+, left panels) and knockout (-/-, right panels) mice at various stages of the first wave, tubular stages are indicated. Sections were counterstained with DAPI. Negative control (secondary antibodies only) is shown (lower panels). Scale bar represents 50 μm. (**B**) Representative pictures of Adgb (green fluorescence) and Sept10 (red fluorescence) in elongating spermatids (stage 12 [S12] upper panels

*Figure 5 continued on next page*

*Figure 5 continued*

and stage 15 [S15] middle panels) after stage-specific tubule dissection of wild-type (+/+) and knockout (-/-) testes. Nuclei were stained with DAPI. Negative controls (secondary antibodies only) are shown (lower panels). Scale bar represents 10 μm. Sperm neck (arrowhead) and annulus (arrow) are highlighted.

The online version of this article includes the following figure supplement(s) for figure 5:

**Figure supplement 1.** Absence of Adgb leads to abnormal annulus migration and mitochondrial disorganization.

conserved GTPases that have the ability to form large oligomers and filamentous polymers and which associate with cell membranes and with the cytoskeleton. They serve as scaffolds for the proper localization of intracellular proteins via their diffusion barrier-forming characteristics (*Dolat et al., 2014*). In sperm, various septins (including Sept1, Sept4, Sept6, Sept7, and Sept12) have been localized to the sperm annulus, where they polymerize to a filamentous structure called the septin ring, forming a barrier between the midpiece and the principal piece of the spermatozoon (*Toure et al., 2011*). Interestingly, despite several members of the septin family being present in the interactome, only Sept10 was found to interact with Adgb in co-IP experiments. Furthermore, the interaction of the two proteins could also be localized within the connecting piece and within the annulus in wild-type sperm, whereas in knockout sperm, only one signal was detected, suggesting the absence of the annulus. Indeed, when staining against Sept7 which interacts with Sept10, the annulus was either missing or failed to migrate properly in knockout sperm, supporting a defect in the annulus formation and migration, thereby also leading to misalignment of the mitochondria in knockout sperm. In accordance with this observation, two main functions have been proposed for the annulus: (1) a diffusion barrier function to compartmentalize different proteins to various locations in the sperm tail and (2) a growth guide function for the sperm flagellum and for aligning the mitochondria along the axoneme (*Avidor-Reiss et al., 2020*; *Avidor-Reiss et al., 2017*; *Toure et al., 2011*). Supportive of this function, *Sept4*$^{-/-}$ and *Sept12*$^{+/-}$ mice are infertile and display disorganized sperm mitochondria (*Ihara et al., 2005*; *Kissel et al., 2005*; *Lin et al., 2009*). Moreover, *Sept4*$^{-/-}$ mice display a bent sperm tail and absence of annulus (*Ihara et al., 2005*; *Kissel et al., 2005*), whereas *Sept12*$^{+/-}$ mice exhibit broken acrosomes, misshaped nuclei, and increased apoptosis of germ cells (*Lin et al., 2009*). Correspondingly, SEPT12 mutations have been described in infertile men displaying abnormal sperm including defective annulus with a bent tail (*Kuo et al., 2012*). Additionally, defective sperm head morphology and DNA integrity have recently been reported for two different SEPT14 missense mutations (*Lin et al., 2020*; *Wang et al., 2019*). Interestingly, another ring-like septin structure was recently described at the sperm neck, composed of Sept12 which complexes together with Sept1, Sept2, Sept10, and Sept11. Two mutations of Sept12 identified in patients disrupted the complex, leading to unstable head-tail junctions and defective connecting piece formation. Strikingly, the mutation of Sept12 and the subsequent disruption of the complex led to loss of Sept10 signal in the annulus (*Shen et al., 2020*) as also observed in Adgb-deficient mice. Accordingly, our data suggest an interdependence between Adgb and Sept10, which is required for the maintenance of the annulus, head shaping, and proper mitochondrial localization.

Sperm flagella and motile cilia, which are hair-like microtubular protrusions at the surface of various cell types, share numerous characteristics, not only in their structure but also in their regulation and growth, whereby septins have also been identified as components of cilia. Sept2 forms a diffusion barrier at the base of the cilium, impeding ciliary formation through loss of Sonic Hedgehog signaling when depleted (*Hu et al., 2010*). Sept2/7/9 form a complex that associates with the ciliary axoneme, thereby regulating ciliary length (*Ghossoub et al., 2013*). Accordingly, septin association with the cytoskeleton and particularly with microtubular structures has been extensively studied (*Spiliotis and Nakos, 2021*), and numerous other cilia-related proteins participate in sperm flagellum formation. Furthermore, most ciliopathies include male infertility and immotile sperm due to defective axonemal organization (*Brown and Witman, 2014*). Likewise, Adgb knockout mice display aberrant microtubule arrangements, thus it cannot be excluded that a scaffolding and simultaneous regulatory action between Adgb and Sept10 is necessary to support microtubular structure. Moreover, a recent study reported the consistent presence of Adgb in the ciliomes of three distinct evolutionary ancestral taxa, further suggesting a conserved function related to microtubular organization and likely flagella formation (*Sigg et al., 2017*). In line with this study, our recent *in vitro* investigations have demonstrated

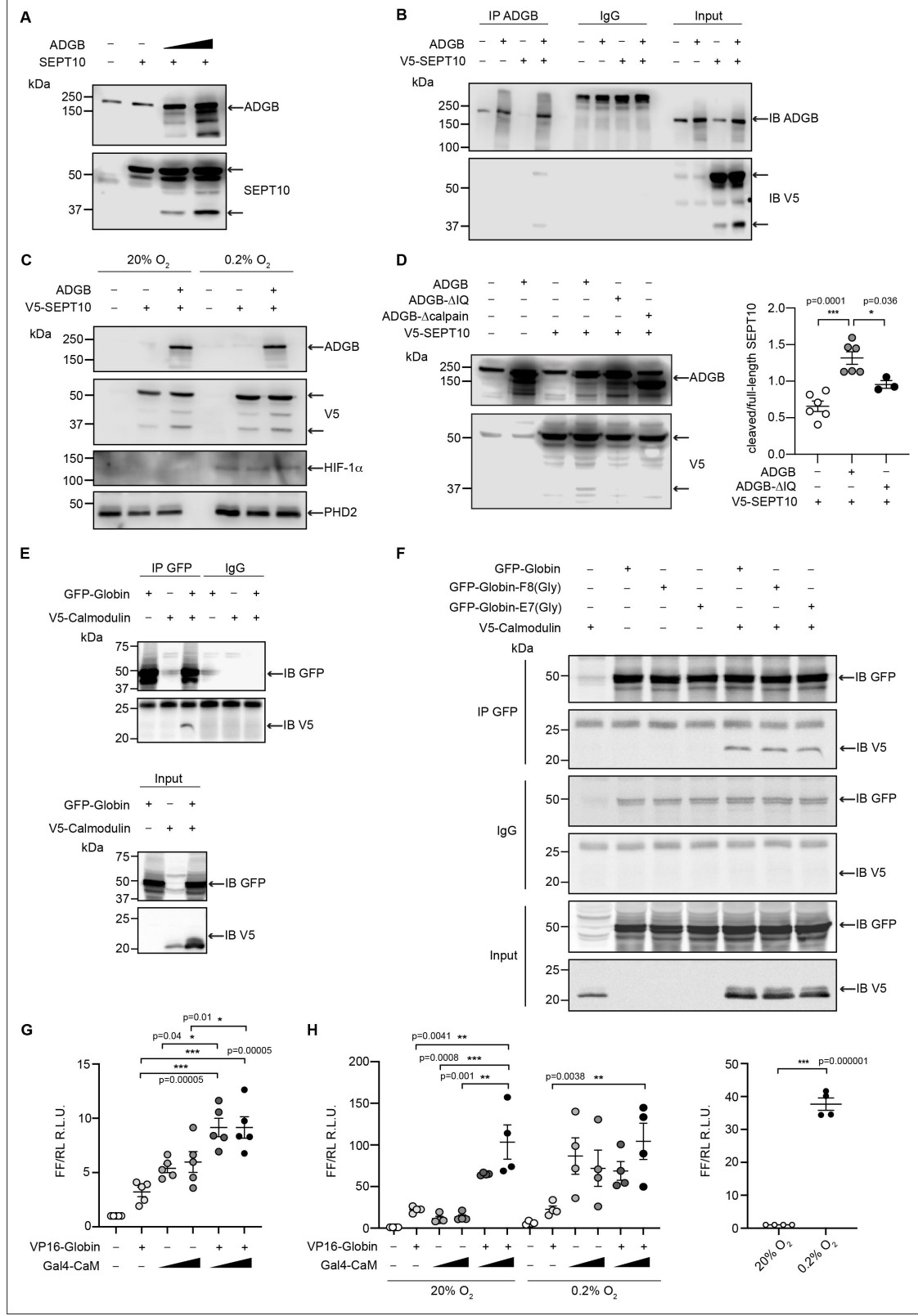

**Figure 6.** ADGB contributes to *in vitro* CaM-dependent SEPT10 cleavage. (**A**) Representative immunoblots of ADGB and SEPT10 in protein lysates of HEK293 cells co-transfected with plasmids encoding SEPT10 and two dose-dependent amounts of full-length ADGB. (**B**) Representative immunoblot of ADGB and V5 in protein lysates of HEK293 cells (co-)transfected with full-length ADGB and a C-terminally V5-tagged SEPT10 construct following co-immunoprecipitation (co-IP) of ADGB and V5-SEPT10. (**C**) Representative immunoblots of ADGB, V5, HIF-1α, and PHD2 in protein lysates of HEK293

*Figure 6 continued on next page*

*Figure 6 continued*

cells (co-)transfected with full-length ADGB and V5-SEPT10 following exposure to normoxic (20% $O_2$) and hypoxic conditions (0.2% $O_2$) for 24 hr. HIF-1α and PHD2 were used as positive controls for hypoxia. (**D**) Representative immunoblot of ADGB and V5 in protein lysates of HEK293 cells (co-)transfected with full-length ADGB, V5-SEPT10, an ADGB-IQ deletion mutant, and an ADGB-calpain protease domain deletion mutant and corresponding protein quantification of cleaved/full-size SEPT10 ratio (n=3–6 independent experiments). Ponceau S protein staining was used as loading control. Schematic representation of deletion constructs is provided in *Figure 4—figure supplement 4*. (**E**) Representative immunoblot of GFP and V5 in protein lysates of HEK293 cells (co-)transfected with a truncated construct of the globin domain of ADGB (spanning the IQ domain) (GFP-Globin) and a V5-tagged CaM (V5-Calmodulin) following IP of GFP. (**F**) Representative immunoblot of GFP and V5 in protein lysates of HEK293 cells (co-)transfected with GFP-Globin, a GFP-Globin construct with mutation of the proximal heme-binding histidine (GFP-Globin-F8[Gly]), a GFP-Globin construct with mutation of the distal glutamine (GFP-Globin-E7[Gly]), and V5-CaM following IP of GFP. (**G, H**) Mammalian 2-hybrid assays in HEK293 cells under normoxic conditions (G) and A375 cells under normoxic and hypoxic (0.2% $O_2$) conditions (H) (n=3–5 independent experiments). HEK293 and A375 cells were transiently transfected with fusion protein vectors based on a Gal4 DNA-binding domain fused to calmodulin (Gal4-CaM) and a VP16 activation domain fused to the ADGB globin domain comprising the IQ domain (VP16-Globin), a Gal4 response element-driven firefly luciferase reporter, and a *Renilla* luciferase control vector. Increasing transfection amounts for the Gal4-CaM fusion protein were employed. Following transfection, A375 cells were incubated under normoxic (20% $O_2$) or hypoxic (0.2% $O_2$) conditions, and luciferase reporter gene activities were determined 24 hr later. Single construct transfections served as negative controls, whereby hypoxic regulation of CaM has been described previously (*Yuan et al., 2005*). A 5'/3'-hypoxia response element-dependent *EPO* promoter-driven firefly luciferase construct served as hypoxic control. * p<0.05, ** p<0.01, and *** p<0.001.

The online version of this article includes the following source data and figure supplement(s) for figure 6:

**Source data 1.** Original uncropped immunoblots of *Figure 6A and B* with indication of the cropped areas.

**Source data 2.** Original uncropped immunoblots of *Figure 6C* with indication of the cropped areas.

**Source data 3.** Original uncropped immunoblots of *Figure 6E* with indication of the cropped areas.

**Source data 4.** Original uncropped immunoblots of *Figure 6F* with indication of the cropped areas.

**Source data 5.** Raw mass spectrometry (MS) data of the ADGB globin IP vs. GFP control IP.

**Figure supplement 1.** Reciprocal co-immunoprecipitation (co-IP) of ADGB and V5-SEPT10 from *Figure 6B*.

**Figure supplement 1—source data 1.** Original uncropped immunoblots of *Figure 6—figure supplement 1* with indication of the cropped areas.

**Figure supplement 2.** Full-length ADGB does not interact with CaM.

**Figure supplement 2—source data 1.** Original uncropped immunoblots of *Figure 6—figure supplement 2* with indication of the cropped areas.

**Figure supplement 3.** ADGB globin immunoprecipitation (IP) vs. GFP control IP.

**Figure supplement 4.** Double mutation of the key heme-binding residues does not abrogate interaction of ADGB and CaM.

**Figure supplement 4—source data 1.** Original uncropped immunoblots of *Figure 6—figure supplement 4* with indication of the cropped areas.

**Figure supplement 5.** Heme depletion does not impact the interaction between GFP-Globin and CaM.

**Figure supplement 5—source data 1.** Original uncropped immunoblots of *Figure 6—figure supplement 5D* with indication of the cropped areas.

**Figure supplement 5—source data 2.** Original uncropped immunoblots of *Figure 6—figure supplement 5E* with indication of the cropped areas.

that Adgb is transcriptionally regulated by FoxJ1, a master regulator of ciliogenesis (*Koay et al., 2021*).

A robust ADGB-SEPT10 interaction was also detected by co-IP experiments in transiently co-transfected HEK293 cells. The interaction persisted despite various mutations of putative binding sites within ADGB, suggesting that the interaction occurs at different locations along the two proteins and that the proteins may entangle around each other. Indeed, it is not uncommon that protein-protein interactions involve multiple interaction surfaces, as referred to in a recent study (*Egri et al., 2022*). Since ADGB contains multiple larger domains of uncharacterized function, it is conceivable that various sites could contribute to interaction with proteins. The close ADGB-SEPT10 interaction may serve as targeting mechanism for the proteolytic processing of SEPT10 that we could observe upon ectopic expression of both proteins. The unique chimeric domain structure of Adgb with an N-terminal calpain-like protease domain and the presence of an IQ motif suggests a CaM-mediated regulation. CaM-bound calcium represents a crucial activator of proteolytic activity of $Ca^{2+}$-dependent calpain proteases (*Bähler and Rhoads, 2002*; *Villalobo et al., 2019*). An interesting observation in our study is the interaction of CaM and ADGB upon isolation of the globin domain but not with the full-length ADGB protein. The lack of interaction upon overexpression of a full-length ADGB might be due to interference by the larger protein (particularly the 700 amino acid long C-terminal region of unidentified origin). As such the globin domain might be liberated possibly by proteolytic cleavage. This hypothesis is consistent with our previous observations of time-dependent truncation

of full-length ADGB from baculovirally infected *Spodoptera frugiperda (Sf9)* insect cells (*Bracke et al., 2018*). Similar truncations are detected upon overexpression of full-length ADGB in HEK293 cells in the current study (e.g. *Figure 6A*). Furthermore, this observation is consistent with the nature of $Ca^{2+}$-dependent calpains, displaying frequent autolytic activation by auto-proteolysis (*Ermolova et al., 2015*; *Osako et al., 2010*). Additional experiments would be required to prove these hypotheses. Importantly, the interaction between CaM and the ADGB IQ-binding motif, as well as presence of the calpain protease domain, seems pivotal in the observed proteolytic cleavage of SEPT10 as the cleavage product completely disappeared upon overexpression of either IQ or protease deletion constructs. Calpain-dependent proteolytic cleavage of septins is not unprecedented; it was shown that Sept5 is a substrate of both calpain-1 and calpain-2 in platelets, where the cleavage triggers secretion of chemokine-containing granules (*Randriamboavonjy et al., 2012*). Septins are sensitive to molecular modifications, such as SUMOylation which impacts Sept6, 7, and 11-dependent filament formation (*Ribet et al., 2017*), or to PKA-dependent phosphorylation as reported for Sept12, leading to its dissociation from the septin complex and disruption of filament formation (*Shen et al., 2017*). It is therefore conceivable that CaM-dependent Adgb-mediated proteolytic cleavage of Sept10 may be a prerequisite for proper Sept10 function or localization within the sperm neck or annulus.

In conclusion, our study is the first to demonstrate a functional role for Adgb, the fifth mammalian globin. We present convincing *in vivo* evidence that Adgb is required for murine spermatogenesis. Adgb is necessary for sperm head shaping and for proper microtubule and flagellum formation. Our *in vitro* data illustrate CaM binding to ADGB and suggest that ADGB contributes to proteolytical cleavage of SEPT10 in a CaM-dependent manner. Our work provides a crucial contribution to the characterization of the physiological role of this novel enigmatic chimeric globin type.

## Materials and methods
### Animals, ethics statement, and genotyping
All experimental procedures and animal maintenance followed Swiss federal guidelines, and the study was revised and approved by the '*Service de la sécurité alimentaire et des affaires vétérinaires*' (SAAV) of the canton of Fribourg, Switzerland (license number 2017_16_FR). Animals were housed in rooms with a 12 hr/12 hr light/dark cycle, controlled temperature and humidity levels and had free access to food and water. Interbreeding of heterozygous animals was performed to obtain wild-type (+/+), heterozygous (Tg/+ for tm1a and tm1b or +/- for tm1d), and homozygous/knockout (Tg/Tg for tm1a and tm1b or -/- for tm1d) littermates that were experimentally used, if not otherwise stated, between 3 and 9 months of age. Genotyping of tm1a and tm1b animals was performed using the following primers (*Figure 1—figure supplement 1*): F1 5'-CCGTGCCCAGCTATATGAGT-3'; R1 5'-CACAACGG GTTCTTCTGTTAGTCC-3'; R2 5'-CCAGCGGTGTTCCTTTCTTA-3'. Primers for tm1d genotyping were the following (*Figure 1—figure supplement 1*): F1, R2, and R4 5'-ACTGATGGCGAGCTCAGACC-3'. PCR amplification was performed for 36 cycles of 1 min at 95°C, 1 min at 56°C, and 1 min at 72°C. The PCR products were separated by electrophoresis on 2% agarose gels and visualized by ethidium bromide staining.

### Gene targeting and knockout mouse generation
The *Adgb*^tm1a(KOMP)Wtsi^ (tm1a) strain was generated by blastocyst microinjection of ESC clone EPD0707_3_ H06, provided by the KOMP (*Skarnes et al., 2011*). Correct targeting of the *Adgb* locus was verified prior to microinjection by long-range PCR using primers 5'S 5'-CTGTACACTGGTTGTACACTGGTA CAACTG-3'; 5'AS 5'-GGACTAACAGAAGAACCCGTTGTG-3'; 3'S 5'-CACACCTCCCCCTGAACCTG AAAC-3'; 3'AS 5'-GTACTTGATTGGACGATGATCCAAG-3' (*Figure 1—figure supplement 1*), generating a band of 6.7 kb for 5' primers and 5.1 kb for 3' primers (*Figure 1—figure supplement 1*). Targeted clones were confirmed by Southern blot analysis using a hybridization probe that targets exon 13 (*Figure 1—figure supplement 1*) revealing a band of 4.1 kb (wild type) or 3.2 kb (tm1a allele) following digestion of genomic DNA with *PvuII* and a band of 2.8 kb (wild type) or 2.4 kb (tm1a allele) following digestion of genomic DNA with *PstI* (*Figure 1—figure supplement 1*). Chimeric mice were bred with C57BL/6-Tyr^c-Brd^ mice, and germline transmission in the F1 offspring was verified by PCR using primers F1, R1, and R2 (*Figure 1—figure supplement 1*). The mice were further bred to C57BL/6N-Hprt^Tg(CMV-cre)Brd/Wtsi^ transgenic mice expressing the Cre allele to delete exons 13 and 14 and

the neo cassette to generate the *Adgb*tm1b(KOMP)Wtsi strain (tm1b), or to C57BL/6N-Gt(ROSA)26Sortm1(FLP1) Dym/Wtsi transgenic mice expressing the Flp recombinase to delete the whole transgene cassette, thereby generating the *Adgb*tm1c(KOMP)Wtsi strain. The latter were further crossed with C57BL/6N-HprtTg(CMV-cre) Brd/Wtsi transgenic mice to delete exons 13 and 14, thereby generating the *Adgb*tm1d(KOMP)Wtsi (tm1d, knockout) strain. The Cre recombinase allele was bred out before any experiments were performed.

## RNA extraction and RT-qPCR

Testes were frozen in liquid nitrogen and stored at –80°C. Tissues were homogenized using a Tissue-Lyser (Qiagen, Valencia, CA, USA). Subsequent RNA isolation and cDNA synthesis were performed as described previously (*Keppner et al., 2019*). In brief, RNA was extracted using an RNeasy Mini Kit (Qiagen) and reverse transcription was performed with 1.5 µg of total RNA and PrimeScript reverse transcriptase (Takara Bio Inc, Kusatsu, Japan). RT-qPCR was performed on a CFX96 C1000 real-time PCR cycler (Bio-Rad Laboratories, Hercules, CA) using SYBRgreen PCR master mix (Kapa Biosystems, London, UK). 21.5 ng of cDNA were loaded, and each sample was run as duplicate. mRNA levels were normalized to β-actin as previously described (*De Backer et al., 2021*). Primer sequences are displayed in *Supplementary file 1*.

## Cell culture and transfection

HEK293 and A375 (ATCC CRL-1619) cells were maintained in Dulbecco's Minimum Essential Media (DMEM) (Gibco, Life Technologies, Carlsbad, CA, USA), containing L-glutamine, supplemented with 10% heat-inactivated fetal bovine serum (FBS; PAN Biotech, Aidenbach, Germany) and 100 Units/ mL penicillin/100 µg/mL streptomycin (Gibco, Life Technologies, Carlsbad, CA, USA). Both cell lines were incubated in a humidified 5% $CO_2$ atmosphere at 37°C and were routinely subcultured after trypsinization. For hypoxic experiments, cells were seeded out in six-well plates or 100 mm culture dishes. On the subsequent day, hypoxia experiments were carried out at 0.2% $O_2$ and 5% $CO_2$ in a gas-controlled glove box (InvivO2 400, Baker Ruskinn, Bridgend, UK) for 24 hr. Transfection of HEK293 cells was performed using calcium-phosphate (*Jordan et al., 1996*), with 2 µg of plasmid DNA for regular immunoblotting experiments and 5 µg of plasmid DNA for immunoprecipitation experiments. Briefly, the DNA was diluted in sterile water and mixed with 250 mM $CaCl_2$. 25 µM chloroquine was added to the cells and allowed to incubate for a minimum of 20 min. Prewarmed 37°C HBS buffer pH 7.05 (NaCl [280 mM], KCl [10 mM], $Na_2HPO_4$ [1.5 mM], D-glucose [12 mM], and HEPES [50 mM]) was added to the DNA solution (50% v/v), and the transfection mixture was added dropwise to the cells. The medium was replaced after 6 hr. Transfection of A375 cells was performed using JetOptimus (Polyplus-transfection SA, Illkirch-Grafffenstaden, France) according to the manufacturer's instructions.

## Heme depletion

To deprive HEK293T cells of intra- and extracellular heme, transfected HEK293T cells were cultured in DMEM medium supplemented with heme-depleted FBS along with 3 mM succinylacetone to inhibit intracellular heme synthesis. Heme depletion in FBS was carried out by incubation with 10 mM ascorbic acid at 37°C for 7 hr, with subsequent dialysis in PBS for three times (*Chen et al., 2012*). This treatment was carried out for 24 hr prior to cell lysis and co-IP.

## SDS-PAGE and immunoblotting

Tissues were homogenized, and proteins for immunoblotting, immunoprecipitation (IP), and proteomics were extracted as described (*Keppner et al., 2019*). Briefly, tissues were harvested and snap-frozen in liquid nitrogen. The tissues were homogenized using a TissueLyser (Qiagen Valencia, CA, USA) in 1 mL protein extraction buffer (Tris-HCl [50 mM], EDTA [1 mM], EGTA [1 mM], sucrose [0.27 mM], leupeptin [2 µg/mL], aprotinin [2 µg/mL], pepstatin [2 µg/mL], PMSF [1 mM], NaF [50 mM], Na-pyro-phosphate [5 mM], and $Na_3VO_4$ [1 mM]). The samples were centrifuged, the supernatant collected, and proteins quantified by Bradford assay. Cells were lysed in triton buffer (Tris-HCl [20 mM, pH 7.4], NaCl [150 mM], and triton X-100 [1%]), left on ice for 15 min, and centrifuged, and the proteins were quantified by Bradford assay.

25 µg of proteins were separated by SDS-PAGE on 10% gels, and proteins were electrotrans-ferred to nitrocellulose membranes (Amersham Hybond-ECL, GE Healthcare, Chicago, IL, USA). The membranes were incubated overnight at 4°C with primary antibody (*Supplementary file 2*) and for

1 hr with donkey anti-rabbit or anti-mouse IgG HRP-conjugated secondary antibody (1:5000, Amersham, Bukinghampshire, UK). All antibodies were diluted in TBS-tween (1%) and dried milk (1%). The signal was revealed using ECL Prime (Amersham, Bukinghampshire, UK) on a C-DiGit Western blot scanner (LI-COR Biosciences) and quantified using ImageStudio program (LI-COR Biosciences, Lincoln, NE, USA). The polyclonal anti-Adgb antibody was custom-made (Proteintech Group Inc, Rosemont, IL, USA). A fusion protein immunogen raised against the 409–745 amino acid region of mouse Adgb was used for the immunization of two rabbits over a period of 102 days. The antibodies in immune sera were affinity purified. Pre-bleeds, test bleeds, and purified antibodies were tested and validated by immunoblotting on wild-type and knockout testis extracts.

## Immunoprecipitation

For immunoprecipitation (IP) of subsequent LC-MS/MS and immunoblotting analyses, 4 and 2 mg of proteins were used, respectively. The protein lysates were first pre-cleared for 24 hr at 4°C with protein G-sepharose beads (GE Healthcare, Chicago, IL, USA) coupled to rabbit IgG (Bethyl Laboratories Inc, Montgomery, TX, USA). Samples were incubated overnight at 4°C with 2 µg primary antibody (*Supplementary file 2*) or 2 µg rabbit IgG, followed by 4 hr with protein G-sepharose beads, then washed two times with wash buffer (Tris-HCl [20 mM, pH 7.4], NaCl [300 mM], and LAP [1 mM]) and three times with equilibration buffer (Tris-HCl [20 mM, pH 7.4], NaCl [150 mM], and LAP [1 mM]). Samples were eluted by boiling for 5 min at 95°C in 2× sample buffer and separated from the beads by centrifugation.

## LC-MS/MS analysis

Washed IP beads were incubated with Laemmli sample buffer (*Laemmli, 1970*), and proteins were reduced with 1 mM DTT for 10 min at 75°C and alkylated using 5.5 mM iodoacetamide for 10 min at room temperature (RT). Protein samples were separated by SDS-PAGE on 4–12% gradient gels (ExpressPlus, Genscript, New Jersey, NJ, USA). Each gel lane was cut into six equal slices, the proteins were in-gel digested with trypsin (Promega, Madison, WI, USA), and the resulting peptide mixtures were processed on StageTips (*Rappsilber et al., 2007*; *Shevchenko et al., 2006*).

LC-MS/MS measurements were performed on a Q Exactive Plus mass spectrometer (Thermo Fisher Scientific, Waltham, MA, USA) coupled to an EASY-nLC 1000 nanoflow HPLC (Thermo Fisher Scientific). HPLC-column tips (fused silica) with 75 µm inner diameter were packed with Reprosil-Pur 120 C18-AQ, 1.9 µm (Dr. Maisch GmbH, Ammerbuch, Germany) to a length of 20 cm. A gradient of solvents A (0.1% formic acid in water) and B (0.1% formic acid in 80% acetonitrile in water) with increasing organic proportion was used for peptide separation (loading of sample with 0% B; separation ramp: from 5 to 30% B within 85 min). The flow rate was 250 nL/min and for sample application 650 nL/min. The mass spectrometer was operated in the data-dependent mode and switched automatically between MS (max. of $1 \times 10^6$ ions) and MS/MS. Each MS scan was followed by a maximum of 10 MS/MS scans using normalized collision energy of 25% and a target value of 1000. Parent ions with a charge state form z=1, and unassigned charge states were excluded from fragmentation. The mass range for MS was m/z=370–1750. The resolution for MS was set to 70,000 and for MS/MS to 17,500. MS parameters were as follows: spray voltage 2.3 kV; no sheath and auxiliary gas flow; ion-transfer tube temperature 250°C. The MS raw data files were uploaded into the MaxQuant software version 1.6.2.10 for peak detection, generation of peak lists of mass error corrected peptides, and for database searches (*Tyanova et al., 2016*). A full-length UniProt mouse (based on UniProt FASTA version April 2016) or human database (UniProt FASTA version March 2016) additionally containing common contaminants, such as keratins and enzymes used for in-gel digestion, was used as reference. Carbamidomethylcysteine was set as fixed modification and protein amino-terminal acetylation and oxidation of methionine were set as variable modifications. Three missed cleavages were allowed, enzyme specificity was trypsin/P, and the MS/MS tolerance was set to 20 ppm. The average mass precision of identified peptides was in general less than 1 ppm after recalibration. Peptide lists were further used by MaxQuant to identify and relatively quantify proteins using the following parameters: peptide and protein false discovery rates (FDRs), based on a forward-reverse database, were set to 0.01, minimum peptide length was set to 7, minimum number of peptides for identification and quantitation of proteins was set to one which must be unique. The 'match-between-run' option (0.7 min) was used.

## Propidium iodide staining and flow cytometry

Preparation of germ cell suspensions was achieved as described (*Jeyaraj et al., 2003*). Briefly, decapsulated testes were incubated in 0.5 mg/mL collagenase type IV in PBS, washed with PBS, and incubated in 1 µg/mL DNase and 1 µg/mL trypsin. Soybean trypsin inhibitor was added, the suspension was filtered, washed in PBS, fixed with 70% ethanol, and stored at 4°C. DNA staining using propidium iodide was performed as described (*Krishnamurthy et al., 2000*). Propidium iodide-stained cells were analyzed in a FACScan flow cytometer (Becton-Dickinson Immunocytometry, San Jose, CA, USA). The fluorescent signals of propidium iodide-stained cells were recorded, and a cytogram of DNA area vs. cell count was used to select cell populations on the basis of their DNA content. A total of 10,000 events was recorded for each histogram. Cell populations were selected based on their DNA content, and their relative numbers were calculated using Summit (Cytomation, CO, USA). Gating was performed as such that quantification represents the ratio of cell size to propidium iodide stainable nuclear DNA content.

## Histological analyses and immunofluorescence

Testes were fixed in 4% paraformaldehyde (PFA) and embedded in paraffin. Preparation of sections and H&E staining was performed as described (*Keppner et al., 2015*). Pictures were taken using a Nikon Eclipse microscope (Nikon Corporation, Tokyo, Japan). For immunofluorescence, testes were fixed in 4% PFA for at least 1 week and incubated in 30% sucrose for another week. The testes were embedded in Optimal Cutting Temperature compound (O.C.T. Tissue-Tek, Sakura Finetek, Tokyo, Japan), and 5 µm thick sections were cut using a cryotome. For seminiferous tubule dissections and stainings, slides were prepared as previously described (*Kotaja et al., 2004*). For sperm stainings, cauda epididymal sperm was retrieved and diluted in PBS. A drop of the suspension was smeared on glass slides and fixed by drying for 15 min and by 4% PFA for 20 min. The slides were blocked in 10% normal goat serum and 0.5% triton X-100 for 1 hr. Testis sections and sperm slides were incubated overnight at 4°C with primary antibodies (*Supplementary file 2*) in 5% normal goat serum and 0.25% triton X-100. The slides were washed with PBS (3 × 10 min), incubated with secondary Alexa Fluor 488 or 594 coupled goat anti-mouse or anti-rabbit IgG (1:300, Invitrogen, Waltham, MA, USA) for 1 hr, washed again with PBS (3 × 10 min), and counterstained with Sudan Black (0.1%) for autofluorescence quenching. Slides were mounted with fluoromount mounting medium containing DAPI (SouthernBiotech, Birmingham, AL, USA) and visualized using a Nikon Eclipse fluorescent microscope (Nikon Corporation).

## *In situ* mRNA hybridization

RNAscope *in situ* hybridization was performed using BaseScope Detection Reagent Kit v2 RED (Advanced Cell Diagnostics Inc, Newark, CA, USA, Cat. No. 323900) according to the manufacturer's instructions. $H_2O_2$ treatment, antigen retrieval, and protease treatment were performed on 5 µm-thick sections prior to hybridization with probes for Adgb (BA-Mm-Adgb-3zz-st, Advanced Cell Diagnostics, Cat. No. 862141), DapB as negative control (BA-DapB-3zz, Advances Cell Diagnostics, Cat. No. 701011), and Ppib as positive control (Ba-Mm-Ppib-3zz, Advanced Cell Diagnostics, Cat. No. 701071) at 40°C for 2 hr followed by eight amplification steps. The signal was revealed with Fast Red, and the sections were counterstained with Gill's hematoxylin no. 1 and mounted with VectaMount (Vector Laboratories, Burlingame, CA, USA). The sections were visualized using a light microscope.

## Electron microscopy

All electron microscopy experiments were performed at the Electron Microscopy Platform of the University of Lausanne, Switzerland. Mouse testes were fixed in 2.5% glutaraldehyde solution (EMS, Hatfield, PA, USA) in phosphate buffer (PB 0.1 M [pH 7.4]) for 1 hr at RT and post-fixed in a fresh mixture of osmium tetroxide 1% (EMS) with 1.5% of potassium ferrocyanide (Sigma, St. Louis, MO, USA) in PB buffer for 1 hr at RT. The samples were then washed twice in distilled water and dehydrated in acetone solution (Sigma) at graded concentrations (30%–40 min; 50%–40 min; 70%–40 min; 100%–2 × 1 hr). This was followed by infiltration in Epon resin (EMS, Hatfield, PA, USA) at graded concentrations (Epon 33% in acetone-4 hr; Epon 66% in acetone-4 hr; Epon 100%–2 × 8 hr) and finally polymerized for 48 hr at 60°C in an oven. Ultrathin sections of 50 nm were cut using a Leica Ultracut (Leica Mikrosysteme GmbH, Vienna, Austria), picked up on a copper slot grid of 2 × 1 mm (EMS,

Hatfield, PA, USA), and coated with a polystyrene film (Sigma, St Louis, MO, USA). Sections were post-stained with uranyl acetate (Sigma, St. Louis, MO, USA) 4% in $H_2O$ for 10 min, rinsed several times with $H_2O$ followed by Reynolds lead citrate in $H_2O$ (Sigma, St Louis, MO, USA) for 10 min, and rinsed several times with $H_2O$. Micrographs were taken with a TEM FEI CM100 (FEI, Eindhoven, The Netherlands) at an acceleration voltage of 80 kV with a TVIPS TemCamF416 digital camera (TVIPS GmbH, Gauting, Germany).

## RNAseq library preparation and transcriptome sequencing

Total RNA from two independent samples of wild-type and Adgb$^{-/-}$ testis was extracted using the mirVana miRNA-Kit according to manufacturer's instructions (Life Technologies, Carlsbad, USA). Prior to library construction, RNA quality was assessed using an Agilent 2100 Bioanalyzer and the Agilent RNA 6000 Nano Kit (Agilent Technologies, Santa Clara, CA, USA). RNA was quantified using Qubit RNA BR Assay Kit (Invitrogen, Waltham, MA, USA). Libraries were prepared starting from 1000 ng of total RNA using the RNA Sample Prep Kit v2 (Illumina Inc, San Diego, CA, USA) including a poly-A selection step following the manufacturer's instructions and sequenced as 2 × 100 nt paired-end reads using an Illumina HiSeq 2500. Library preparation and sequencing were performed by the NGS Core Facility of the Department of Biology, Johannes-Gutenberg University (Mainz, Germany). RNAseq data are available from the European Nucleotide Archive under accession number PRJEB46499.

## Differential gene expression, GO term annotation, and pathway enrichment analyses

Raw sequences were pre-processed to remove low quality reads and residual Illumina adapter sequences using BBduk from the BBtools suite (https://sourceforge.net/projects/bbmap/). The overall sequencing quality and the absence of adapter contamination were evaluated with FastQC. Mapping was performed with HISAT2, and quantification of gene expression was done using StringTie. Differentially expressed genes were determined using DESeq2. Genes were considered differentially expressed when presenting |fold change|>2 and FDR-corrected p-value≤0.1. GO term enrichment analyses were performed using WebGestalt 2019 using the Overrepresentation Enrichment Analysis method, requiring a BH-corrected p-value≤0.05 and a minimum enrichment of four genes for term/ pathway. Enrichment in canonical pathways were performed with Qiagen's Ingenuity Pathway Analysis (IPA, Qiagen, Hilden, Germany), Core analysis tool using bias-corrected z-score (when applicable), and BH-corrected p-values≤0.05.

## Cloning and construction of expression plasmids

Generation of pLenti6-ADGB was described before (*Bracke et al., 2018*), pLenti6-SEPT10-V5 was purchased from DNASU (clone ID HsCD00943271, DNASU Plasmid Repository, Arizona State University, AZ, USA). All additional recombinant genes were cloned into pFLAG-CMV™-6a expression vector (Sigma) unless otherwise specified. All coding sequences were amplified by PCR using Phusion High-Fidelity DNA polymerase (Thermo Fisher Scientific). Recombinant *ADGB*, *SEPT10,* and *SEPT7* with N-terminal FLAG tag and C-terminal myc tag were constructed by amplifying and ligating their respective coding sequence with in-primer designed myc tag (EQKLISEEDL) into the expression vector, in-frame with the N-terminal FLAG tag. A glycine-serine (GSG) linker was added between the last codon of *SEPT10* and the first codon of the myc tag. *SEPT2*, *SEPT11,* and *SEPT12* expression vectors were cloned by amplifying and ligating their respective coding sequence into the pFLAG-CMV-6a expression vector containing an N-terminal FLAG tag. Truncated ADGB proteins consisting of the calpain-like domain, 350-residue uncharacterized domain and globin domain (N-terminal mutant), or the 700-residue uncharacterized region (C-terminal mutant) domain were designed with GFP tags at both N- and C-termini. The N-terminal mutant ADGB was amplified between codons of residues 58 and 968 in ADGB, while the C-terminal mutant ADGB was amplified between codons of residues 969 and 1667. Amplicons were designed with 5'- and 3'- overhangs compatible with two customized GFP amplicons designed to anneal at the 5'- and 3'-ends of the genes. GSG linkers (GGSGGGGSGG) were added to bridge the GFP tags and the truncated ADGB proteins. Similarly, N-terminally GFP-tagged isolated ADGB globin domain was cloned by amplifying and ligating amplicons of the *ADGB* globin coding sequence downstream to a *GFP* coding sequence with complementary overhangs, with the GSG linker added between the two proteins. With the same construction, GFP-tagged ADGB

globin domains with a single mutation on the proximal histidine in helix F codon 8 (H824G) or the distal glutamine in helix E codon 7 (Q792G), or both (H824G/Q792G) were cloned by amplifying and ligating the globin domain coding sequence using primers designed to carry the mutated codon sequence. ADGB-ΔIQ, ADGB-Δcalpain, and ADGB-ΔCCD were constructed by amplifying designed *ADGB* amplicons with compatible overhangs and were ligated in-frame to generate an *ADGB* coding sequence with deletions in the desired domains. For M2H assays, Gal4-CaM, Gal4-ADGB, VP16-globin domain, VP16-CaM, or VP16-ADGB were cloned into a pcDNA3.0 expression vector. Gal4 DNA-binding domain or VP16 transactivation domain sequences were amplified with the in-primer designed GSG linker at the 3'-end of the amplicons. The coding sequence of *CALM3*, *ADGB* globin domain, and *ADGB* full length were amplified with complementary 5'-end and ligated to the Gal4 and VP16 sequences to generate the fusion genes.

## Reporter gene assays

For M2H assays, $2.15 \times 10^5$ HEK293 or $4 \times 10^5$ A375 cells were transiently transfected with 1 µg firefly luciferase reporter plasmid (5xGAL4-TATA-luciferase, Addgene, 46756) (*Sun et al., 1994*) and 500 ng or 200 and 300 ng chimeric Gal4 and VP16 fusion protein vectors, respectively, in 12- or 6-well format using $CaCl_2$ or JetOptimus. To control for differences in transfection efficiency and extract preparation, 25 ng or 50 ng pRL-SV40 *Renilla* luciferase reporter vector (Promega, Madison, WI, USA) was co-transfected, respectively for HEK293 and A375 cells. Cultures were evenly split onto 12-well plates 24 hr after transfection for A375 cells. For hypoxia control experiments, $4 \times 10^5$ A375 cells were transiently co-transfected with 500 ng firefly 5'/3'-hypoxia response element-dependent *EPO* promoter-driven luciferase reporter plasmid (*Storti et al., 2014*) and 50 ng pRL-SV40 *Renilla* luciferase reporter vector. Luciferase activities of duplicate wells were determined using the Dual Luciferase Reporter Assay System (Promega) as described before (*Schörg et al., 2015*). Reporter activities were expressed as relative firefly/*Renilla* luciferase activities. All reporter gene assays were performed at least three times independently.

## Testosterone quantification

For intra-testis testosterone quantification, testis samples were homogenized twice using the Precellys 24 tissue homogenizer (Bertin Instruments; Rockville, MD, USA) (4°C, 3×, 30 s at 6500 rpm, cycle break 30 s) in chloroform-isopropanol (1 mL, 50/50%; v/v) containing ISTD. Combined supernatants were centrifuged (10 min, RT, $16,000 \times g$) and evaporated using a Genevac EZ-2 evaporator (Stepbios, Muttenz, Switzerland) (3 hr, 35°C). Samples were reconstituted in methanol (50 µL, 10 min, RT, 1300 rpm) and sonicated (10 min, RT). Reconstituted samples were centrifuged (10 min, RT, $16,000 \times g$), and supernatants were transferred to LC-MS vials. Testosterone content was analyzed by ultra-performance liquid chromatography-MS/MS (UPLC-MS/MS) using an Agilent 1290 Infinity II UPLC coupled to an Agilent 6495 triple quadrupole mass spectrometer equipped with a jet-stream electrospray ionization interface (Agilent Technologies). Analyte separation was achieved using a reverse-phase column (1.7 µm, 2.1 mm × 150 mm; Acquity UPLC BEH C18; Waters). Data acquisition and quantitative analysis were performed by MassHunter (Version B.10.0. Build 10.0.27, Agilent Technologies).

## TUNEL assay

TUNEL assay was performed on paraffin-embedded testis sections using the DeadEnd Fluorometric TUNEL System according to manufacturer's instructions (Promega, Madison, WI, USA). The positive control sections were pre-incubated with DNase I solution (Qiagen, Valencia, CA, USA), while negative control sections were incubated in incubation buffer without recombinant terminal deoxynucleotidyl transferase (rTdT enzyme). Slides were mounted with fluoromount mounting medium containing DAPI (SouthernBiotech, Birmingham, AL, USA) and visualized using a Nikon Eclipse fluorescent microscope (Nikon Corporation). 10 pictures were taken for each animal and TUNEL-positive cells were counted using FIJI software (*Schindelin et al., 2012*).

## *ELISA* quantification of LH and FSH

Serum LH and FSH were quantified by ELISA according to manufacturer's instructions using 50 µL of sample (Cloud-Clone Corp., Houston, TX, USA).

## Statistical analysis

All values were presented as mean ± SEM. Differences in means between two groups were analyzed with unpaired two-tailed Student's t-test (*Figure 1A, C, D and E*; *Figure 6H* right graph; *Figure 1—figure supplement 2A,B Figure 1—figure supplement 3A-G,I Figure 4—figure supplement 1H-N Figure 6—figure supplement 5A,B,C*) and those among multiple groups with one-way ANOVA followed by Tukey post-hoc test (*Figure 2D*; *Figure 6D, G and H* left graph, *Figure 6—figure supplement 2B*). All statistics were performed with GraphPad Prism software 7.05. Values of p≤0.05 were considered statistically significant.

## Acknowledgements

We thank Christine Roulin for technical assistance. We thank Damien De Bellis from the Electron Microscopy Platform of the University of Lausanne for EM section preparation and image acquisition. This work was supported by the Swiss National Science Foundation to DH (grants 31,003 A_173000 and 310030_207460) and the German Research Foundation to DH (HO 5837/1–1) and TH (HA 2103/9–1).

## Additional information

### Funding

| Funder | Grant reference number | Author |
|---|---|---|
| Schweizerischer Nationalfonds zur Förderung der Wissenschaftlichen Forschung | 31003A_173000 | David Hoogewijs |
| Schweizerischer Nationalfonds zur Förderung der Wissenschaftlichen Forschung | 310030_207460 | David Hoogewijs |
| Deutsche Forschungsgemeinschaft | HO 5837/1-1 | David Hoogewijs |
| Deutsche Forschungsgemeinschaft | HA 2103/9-1 | Thomas Hankeln |

The funders had no role in study design, data collection and interpretation, or the decision to submit the work for publication.

### Author contributions

Anna Keppner, Conceptualization, Data curation, Formal analysis, Investigation, Methodology, Visualization, Writing – original draft; Miguel Correia, Formal analysis, Investigation, Visualization; Sara Santambrogio, Teng Wei Koay, Darko Maric, Denise V Winter, Frédéric Chalmel, Formal analysis, Investigation, Methodology; Carina Osterhof, Formal analysis, Investigation, Methodology, Visualization; Angèle Clerc, Investigation; Michael Stumpe, Dieter Kressler, Data curation, Formal analysis, Methodology; Sylvia Dewilde, Formal analysis; Alex Odermatt, Formal analysis, Methodology, Resources; Thomas Hankeln, Data curation, Formal analysis, Funding acquisition, Methodology; Roland H Wenger, Formal analysis, Investigation, Resources, Writing - review and editing; David Hoogewijs, Conceptualization, Data curation, Formal analysis, Funding acquisition, Investigation, Methodology, Project administration, Supervision, Writing – original draft

### Author ORCIDs

Carina Osterhof ⓘ http://orcid.org/0000-0002-1699-7410
Michael Stumpe ⓘ http://orcid.org/0000-0002-9443-9326
Dieter Kressler ⓘ http://orcid.org/0000-0003-4855-3563
David Hoogewijs ⓘ http://orcid.org/0000-0001-5547-6004

### Ethics

All experimental procedures and animal maintenance followed Swiss federal guidelines and the study was revised and approved by the "Service de la sécurité alimentaire et des affaires vétérinaires"(SAAV) of the canton of Fribourg, Switzerland (license number 2017_16_FR).

### Decision letter and Author response

Decision letter https://doi.org/10.7554/eLife.72374.sa1
Author response https://doi.org/10.7554/eLife.72374.sa2

## Additional files

### Supplementary files
- Transparent reporting form
- Supplementary file 1. List of primers used for RT-qPCR.
- Supplementary file 2. List of antibodies used throughout the study.

### Data availability

RNA-sequencing data have been submitted to ENA with accession number PRJEB46499 and is also available as supplemental dataset 1 (excel table). All data generated or analysed during this study are included in the manuscript and supporting files. Source data files are provided for Figures 1, 2, 4B, 4C, 4D, 4E, 6A, 6B, 6C, 6D, 6E, 6F, Fig. 1-fig. suppl. 1, Fig. 4-fig. suppl. 1A, B, C, D, E, F, G, Fig. 4-fig. suppl. 2A, B, C, Fig. 4-fig. suppl. 3A, B, C, D, E, Fig. 4-fig. suppl. 5, Fig. 4-fig. suppl. 6B, Fig. 6-fig. suppl. 1A, Fig. 6-fig. suppl. 4, Fig. 6-fig. suppl. 5D, E.

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
