## [Editor Report]

This manuscript demonstrates that male mice lacking androglobin, a poorly understood heme-containing protein, are infertile and have defects in late stage spermatogenesis. The revisions are thorough and the inclusion of additional data makes the manuscript solid. The lack of defects at the hypothalamus-pituitary level makes the phenotype more striking as a direct effect of the mutation at the testis level. Overall, this revised version is significantly improved.

---

## [Decision Letter]

**Decision letter after peer review:**

Thank you for submitting your article "Androglobin, a chimeric mammalian globin, is required for male fertility" for consideration by *eLife*. Your article has been reviewed by 3 peer reviewers, and the evaluation has been overseen by a Reviewing Editor and Anna Akhmanova as the Senior Editor. The following individual involved in review of your submission has agreed to reveal their identity: Marie-Alda Gilles-Gonzales (Reviewer #1).

Summary:

Although generation and characterization of androglobin null mice and identification of androglobin interacting partners are major strengths of this interesting manuscript, several concerns were raised as described below. The distinction between primary versus secondary effects of loss of andrglobin are not well-detailed, the global knockout model could have defects in the hypothalamus and pituitary or even testis levels and these may result in secondary phenotypes, sperm flagellar structural integrity in term so microtubule arrangements has not been described and the rationale for many of the experiments is not clearly described.

Essential revisions:

1. Rewrite introduction in response to Rev 2 comments.

2. Distinguish primary versus secondary effects- hormone assays need to be performed to measure serum levels of gonadotropins.

3. PAS staining of mutant testes needs to be done to identify spermatogenesis stages.

4. TUNEL assays are needed to demonstrate apoptosis.

5. Sperm flagellar structure need to be analyzed.

6. Immunofluorescence data on intact sections are required.

7. The data on delete constructs is inconclusive. Expression of other interacting proteins need to be evaluated.

8. The lack of interaction of full length androglobin protein needs to be better explained.

*Reviewer #1 (Recommendations for the authors):*

Apart from a nod to the discoverer of the globin-coupled sensors, the late Dr. Maqsudul Alam, which I think is important here, the paper may be published as is.

The ligand affinity of hexacoordinate heme may be allosterically influenced by small molecules, or by interacting proteins. Therefore, it might be worthwhile to examine, in future, the effect of low-molecular weight testes extracts, and other interacting proteins, on a possible O2-dependent in vitro cleavage of Sept10.

*Reviewer #2 (Recommendations for the authors):*

1. Page 2, lines 21-22: Hundreds, if not thousands, of genes have been identified to cause sperm deformation and male infertility. Why did the authors think it is "unknown"?

2. Th introduction is a disaster as if it was written by a layperson who does not know much about spermatogenesis and male fertility. Almost all statements in this section are inaccurate or maybe totally wrong. References are not accurately cited, and the main messages of those papers cited were mostly misinterpreted. Here are some examples: (1) lines 40-41: logically confusing? Sperm elongation is part of haploid cell differentiation/spermiogenesis, which is also part of the male germ cell differentiation/spermatogenesis. The authors need to consult an expert on spermatogenesis. (2) Lines 42-44: SSC renewal is not like what is described here!! Again, true experts need to be consulted. (3) Lines 44-45: Wrong! Spermatogonia proliferate to multiply themselves. Primary spermatocytes are derived from type B spermatogonia when the meiotic program is turned on. (4) Lines 47-49: This is completely wrong! The chromatoid body (CB) only represents an RNA processing center. Where did you get the idea that CBs stored materials required for spermatid development? Spermatid maturation is a wrong term. (5) Lines 50-53: Spermiogenesis is a spermatid differentiation process. Maturation is a wrong term. (6) Lines 55-56: How do "these different steps lead to male fertility"? Lines 59-61: Thousands of genes have been identified to be essential for the three phases of spermatogenesis. Why do you say these are all "unknown"? To an investigator working on spermatogenesis for decades, this is beyond insulting! (7) Lines 87-88: It sounds like the events in spermiogenesis are following some orders or sequential. This is simply not true. The introduction must be rewritten completely.

3. Page 7, line 115: Decreased testosterone levels are very rare in KOs with only disruptions in late spermiogenesis. This finding suggests that the HPT axis is messed up probably due to defects in the hypothalamus and/or pituitary in the global Adgb KO mice. For this reason, levels of FSH, LH, and testicular T levels must be measured and presented. If brain defects are identified, then the testicular/sperm phenotype may represent secondary effects; therefore, a conditional KO line is needed to exclude the contributions from brain defects in the global KO mice.

4. Lines 116-123: PAS staining is needed to accurately determine the stages of the seminiferous epithelial cycle. In Figure 1F, WT and KO sections are not from the same stages and thus, not comparable at all. What is a cell bulge?? The terminologies used are non-standard.

5. Lines 126-129: Why not using TUNEL to detect apoptotic germ cells? Dysregulation of apoptosis-related mRNAs does not mean apoptosis.

6. Lines 149-155: Based on the stages, one can accurately map the expression of Adgb mRNA and protein to specific cell types in the seminiferous epithelium. By doing so, the onset of mRNA and protein can be determined. The delayed translation is an important feature of gene regulation during spermiogenesis, and this gene apparently displays a delayed translation.

7. Lines 167-169: Since this gene has been shown to be implicated in ciliogenesis, the focus should be put on sperm flagella formation. Is the 9+2 microtubule structure normal in the KO sperm? How about outer dense fibers? Since sperm tails start to form at step 9, the TEM analyses should be focused on tubules from stages IX and X. The sections from WT and KO were not from the same stages and thus, non-comparable!!!

8. Lines 183-185: Is this gene expressed in Leydig cells? How come disruptions of elongated spermatids cause decreased levels of T?

9. Lines 215-216: Immunofluorescence should be performed using intact testis sections, not isolated elongating spermatids because the structural integratory and cellular context are all lost.

10. Discussion is too long and unfocused. The most important problems were not discussed, e.g., what disruptions are specific to Adgb ablation? What are the primary defects? And what are the secondary defects? how does ablation Adgb lead to the structural defects observed? If through Septins, then how?

11. Lines 415-416: Sept10 KO is needed to draw this conclusion.

*Reviewer #3 (Recommendations for the authors):*

The manuscript of Keppner et al. is a novel preliminary characterization of a testes specific noncanonical globin protein called Androglobin. This work builds off of the authors prior findings over the last decade on the androglobin gene, generating the first Adgb knockout mouse model to explore its physiological functions. Herein they convincingly demonstrate that the loss of Adgb results a striking reproductive phenotype, where knockout male mice are infertile and azoospermic due to complete disruption of spermatid elongation and maturation. They attempt to explain how this phenotype may manifest in these mice and show that Adgb binds with a number of critical proteins for spermatogenesis including septins. However, the only mechanism they are able to implicate (primarily via in vitro over expression studies) is that Adgb may be capable of proteolytic cleavage of septins via a calmodulin dependent interaction. Though certainly an exciting theory, the reasoning and mechanisms behind these phenomena are unexplored and warrant further investigation.

Lines 193-200. The rationale for Adgb interaction specifically with Sept 10 is not provided. Does Adgb interact with the other Sept (2,7,11) found in the IP proteomics? In Line 209, what was the point of the deletion constructs because at the end all 3 domain deletions still interacted. So we are back to square-one. Was CaM identified in the proteomic experiments in the testes?

Line 171-185 RNAseq section – Specific GO terms p values or rank output for IPA analysis are not shown – these need to be included. At a minimum the top 10 GO terms from IPA with their consensus fold direction would be highly informative. Would also be illustrative and helpful to the reader for the authors to label and highlight individual genes with the largest fold changes that are hyper significant in the figure S1A volcano plot.

Line 194-195 The rational for Sept10 is a pretty weak statement, what about other proteins like Tprn or Mkl2 which are more specific and abundant? Even Spef2 (sperm flagellar protein) would have been much more interesting given phenotype observed. Authors should either elaborate further or modify.

Lines 214-227 For IFs of spermatids figure 5, in general localization of Adgb from images is difficult to appreciate, appears to be a large amount of unspecific staining. At a minimum, KO spermatids should be used as a negative control for early stage (s7, 12) as they are still present and will help clarify (2nd only panel is not sufficient, will never see any signal and you have not shown those same stages). Additionally, subpanel B does not enhance the figure substantially given that the sperm are already unhappy at this point. Perhaps show CoxIV staining at earlier stages than mature epididymal sperm.

Line 240 The introduction of calmodulin requires more explanation or background, it is only mentioned in the introduction when discussing Adgb's structure once as a putative motif and then becomes a major focus for the rest of the manuscript and discussion.

Of note, Figure 6D is one of the most striking pieces of data in the entire manuscript, very nice! But how does ADGB perform this function? Does this occur through its N-terminal protease domain?

Lines 244-247 Why does the full length not interact and why is the data not show? Authors should comment on this further, additionally the data shown from the interaction with the truncated globin gfp is not convincing that this is a "robust" interaction and needs to be supported by the expression of V5 calmodulin alone control which is absent from figure 6E.

Lines 250-252 Experiments from figures 6F and S10 are nicely done but what happens when you modify heme content? Deplete heme from the media or block heme synthesis? How do you know these mutant truncated globins aren't still hemylated? And what happens when you make these same mutations in the full length Adgb? All these results demonstrate is that these specific constructs still interact in vitro… it is not indicative that heme coordination is expendable.

Lines 259-263 and figure 6 legend describing the mammalian-2 hybrid assays under normoxic or hypoxic conditions – the authors claim exposure to hypoxia does not alter the luciferase reporters. However, in the legend they state that they do not include the Gal4-CaM alone control condition because of its hypoxic regulation. How then is this a good model system? And how do the authors explain the single transfection alone being modulated by hypoxia but the dual expression not? If anything, it is an essential control, particularly in comparison to the Epo construct they have used as a hypoxic positive control. This data should not be omitted and should be shown and explained/interpreted.

---

## [Author Response]

Essential revisions:1. Rewrite introduction in response to Rev 2 comments.

We have fully rewritten the introduction.

2. Distinguish primary versus secondary effects- hormone assays need to be performed to measure serum levels of gonadotropins.

We performed hormone assays.

3. PAS staining of mutant testes needs to be done to identify spermatogenesis stages.

We performed PAS stainings.

4. TUNEL assays are needed to demonstrate apoptosis.

We performed TUNEL assays.

5. Sperm flagellar structure need to be analyzed.

We performed novel EM analyses to describe sperm flagellar structure.

6. Immunofluorescence data on intact sections are required.

We performed novel immunofluorescence experiments on intact sections, including KO sections as negative control.

7. The data on delete constructs is inconclusive. Expression of other interacting proteins need to be evaluated.

We have performed additional co-IP experiments and analyzed interaction in lysates as well as upon overexpression with several additional proteins. We have amended the discussion on the deletion constructs.

8. The lack of interaction of full length androglobin protein needs to be better explained.

We extensively explained the lack of interaction of full length androglobin protein with calmodulin upon overexpression in the discussion.

Reviewer #1 (Recommendations for the authors):Apart from a nod to the discoverer of the globin-coupled sensors, the late Dr. Maqsudul Alam, which I think is important here, the paper may be published as is.

We appreciate the reviewer’s very positive evaluation of our manuscript and are thankful for the suggestion. We have added references to Maqsudul Alam’s work on globin-coupled sensors in the introduction.

The ligand affinity of hexacoordinate heme may be allosterically influenced by small molecules, or by interacting proteins. Therefore, it might be worthwhile to examine, in future, the effect of low-molecular weight testes extracts, and other interacting proteins, on a possible O2-dependent in vitro cleavage of Sept10.

We thank the reviewer for this suggestion.

Reviewer #2 (Recommendations for the authors):1. Page 2, lines 21-22: Hundreds, if not thousands, of genes have been identified to cause sperm deformation and male infertility. Why did the authors think it is "unknown"?

We have changed the sentence.

2. The introduction is a disaster as if it was written by a layperson who does not know much about spermatogenesis and male fertility. Almost all statements in this section are inaccurate or maybe totally wrong. References are not accurately cited, and the main messages of those papers cited were mostly misinterpreted. Here are some examples: (1) lines 40-41: logically confusing? Sperm elongation is part of haploid cell differentiation/spermiogenesis, which is also part of the male germ cell differentiation/spermatogenesis. The authors need to consult an expert on spermatogenesis. (2) Lines 42-44: SSC renewal is not like what is described here!! Again, true experts need to be consulted. (3) Lines 44-45: Wrong! Spermatogonia proliferate to multiply themselves. Primary spermatocytes are derived from type B spermatogonia when the meiotic program is turned on. (4) Lines 47-49: This is completely wrong! The chromatoid body (CB) only represents an RNA processing center. Where did you get the idea that CBs stored materials required for spermatid development? Spermatid maturation is a wrong term. (5) Lines 50-53: Spermiogenesis is a spermatid differentiation process. Maturation is a wrong term. (6) Lines 55-56: How do "these different steps lead to male fertility"? Lines 59-61: Thousands of genes have been identified to be essential for the three phases of spermatogenesis. Why do you say these are all "unknown"? To an investigator working on spermatogenesis for decades, this is beyond insulting! (7) Lines 87-88: It sounds like the events in spermiogenesis are following some orders or sequential. This is simply not true. The introduction must be rewritten completely.

We apologize if our wording provoked any insult. We have entirely revised the introduction and particularly paid attention to all remarks raised by the reviewer. According to the suggestion of the reviewer we have consulted an expert to accompany us (F.C.). Given the suggestion of reviewer 3 on calmodulin we have enlarged the introduction on the link between calmodulin and spermatogenesis and in turn written the spermatogenesis paragraph more concisely.

3. Page 7, line 115: Decreased testosterone levels are very rare in KOs with only disruptions in late spermiogenesis. This finding suggests that the HPT axis is messed up probably due to defects in the hypothalamus and/or pituitary in the global Adgb KO mice. For this reason, levels of FSH, LH, and testicular T levels must be measured and presented. If brain defects are identified, then the testicular/sperm phenotype may represent secondary effects; therefore, a conditional KO line is needed to exclude the contributions from brain defects in the global KO mice.

We have measured serum levels of FSH, LH, and testicular testosterone levels. The results are integrated as Figure 1E (testicular testosterone) and Figure 1-Figure suppl. 2 for FSH and LH. No brain defects were identified, suggesting a primary sperm phenotype. Accordingly, we have added a paragraph referring to this in the discussion.

4. Lines 116-123: PAS staining is needed to accurately determine the stages of the seminiferous epithelial cycle. In Figure 1F, WT and KO sections are not from the same stages and thus, not comparable at all. What is a cell bulge?? The terminologies used are non-standard.

We have performed PAS stainings. The results are integrated as Figure 1F. We reworded the non-standard terminology.

5. Lines 126-129: Why not using TUNEL to detect apoptotic germ cells? Dysregulation of apoptosis-related mRNAs does not mean apoptosis.

We have performed TUNEL assays. The results are integrated as Figure 1-Figure suppl. 3H and Figure 1-Figure suppl. 3I.

6. Lines 149-155: Based on the stages, one can accurately map the expression of Adgb mRNA and protein to specific cell types in the seminiferous epithelium. By doing so, the onset of mRNA and protein can be determined. The delayed translation is an important feature of gene regulation during spermiogenesis, and this gene apparently displays a delayed translation.

We have alluded to the delayed translation in the text.

7. Lines 167-169: Since this gene has been shown to be implicated in ciliogenesis, the focus should be put on sperm flagella formation. Is the 9+2 microtubule structure normal in the KO sperm? How about outer dense fibers? Since sperm tails start to form at step 9, the TEM analyses should be focused on tubules from stages IX and X. The sections from WT and KO were not from the same stages and thus, non-comparable!!!

We have performed additional TEM analyses of stage-matched samples and entirely revised the figure. These results are incorporated in Figures 3C, 3D, 3E, 3F.

8. Lines 183-185: Is this gene expressed in Leydig cells? How come disruptions of elongated spermatids cause decreased levels of T?

Given the novel analyses based on testicular testosterone, we have removed the previous serum testosterone measurements. Indeed, we find no convincing expression in Leydig cells as we have already stated in the discussion of the initially submitted manuscript. The newly included more recent scRNAseq data (Figure 2-Figure suppl. 2) confirmed the absence of Adgb mRNA in Leydig cells. We have referred to this now in the Discussion section.

9. Lines 215-216: Immunofluorescence should be performed using intact testis sections, not isolated elongating spermatids because the structural integratory and cellular context are all lost.

Immunofluorescence experiments were performed on intact testis sections and integrated in Figure 5.

10. Discussion is too long and unfocused. The most important problems were not discussed, e.g., what disruptions are specific to Adgb ablation? What are the primary defects? And what are the secondary defects? how does ablation Adgb lead to the structural defects observed? If through Septins, then how?

We have substantially shortened the discussion and focused it more with integration of the novel data.

11. Lines 415-416: Sept10 KO is needed to draw this conclusion.

We have reformulated the sentence (also in abstract).

Reviewer #3 (Recommendations for the authors):The manuscript of Keppner et al. is a novel preliminary characterization of a testes specific noncanonical globin protein called Androglobin. This work builds off of the authors prior findings over the last decade on the androglobin gene, generating the first Adgb knockout mouse model to explore its physiological functions. Herein they convincingly demonstrate that the loss of Adgb results a striking reproductive phenotype, where knockout male mice are infertile and azoospermic due to complete disruption of spermatid elongation and maturation. They attempt to explain how this phenotype may manifest in these mice and show that Adgb binds with a number of critical proteins for spermatogenesis including septins. However, the only mechanism they are able to implicate (primarily via in vitro over expression studies) is that Adgb may be capable of proteolytic cleavage of septins via a calmodulin dependent interaction. Though certainly an exciting theory, the reasoning and mechanisms behind these phenomena are unexplored and warrant further investigation.Lines 193-200. The rationale for Adgb interaction specifically with Sept 10 is not provided. Does Adgb interact with the other Sept (2,7,11) found in the IP proteomics? In Line 209, what was the point of the deletion constructs because at the end all 3 domain deletions still interacted. So we are back to square-one. Was CaM identified in the proteomic experiments in the testes?

We now provide the rationale for Sept10 substantiated by additional experiments outlined below: while among the specifically enriched proteins in the IP proteomics experiment, multiple members of the septin family of proteins were present (Sept10, Sept11, Sept2, and Sept7), Sept10 was outstanding given its strong enrichment combined with its high abundance in the immunoprecipitation.

We have performed additional reciprocal co-IP experiments in testis lysates for Sept2, Sept7 and Sept11 using septin-specific antibodies (Figure 4-Figure suppl. 2). We have generated novel plasmids for SEPT2, SEPT7 as well as SEPT11 and performed reciprocal co-IP experiments following transfection and overexpression of these constructs. We also included SEPT12 in these assays as SEPT12 complexes together with SEPT7 and SEPT2, and SEPT12 have been shown to be crucial for septin ring/sperm annulus formation (Kuo et al., 2015, *J. Cell Sci.* 128, 923–934; Shen et al., 2017, *PLoS Genet.* 13(3): e1006631). Whereas these experiments illustrate lack of interaction between Adgb and all septins other than Sept10, we could convincingly confirm the interaction between SEPT7 and SEPT10 by reciprocal co-IP, further substantiating that Sept10 is the prime interactor of Adgb (Figure 4-Figure suppl. 3). These additional experiments argue as well in favor of proceeding with Sept10 for downstream experiments.

Multiple Adgb deletion constructs were used to delineate the interaction domain. However, instead of omitting these data, we prefer to fully report them. We believe that these findings underscore the robustness of the interaction, possibly via the multiple large domains of uncharacterized function present in ADGB. Indeed, it is not so uncommon that protein-protein interactions involve multiple interaction surfaces. Moreover, the finding that ADGB interacts both via its N- and C-terminal part with SEPT10 (Figures 4D, E) is an interesting observation that may possibly be of mechanistic relevance. Exploring the functional significance of this complex interaction pattern will be the subject of a follow-up study.

CaM was identified by MS in testis lysates following ADGB co-IP, but only at very low enrichment over the control. However, endogenous CaM was more robustly detected by MS in HEK293 cells following overexpression of the ADGB globin domain. This finding is consistent with the detection of exogenous CaM binding to the overexpressed globin domain only, liberated from the full-length protein impeding the interaction between CaM and ADGB, as referred to below.

Line 171-185 RNAseq section – Specific GO terms p values or rank output for IPA analysis are not shown – these need to be included. At a minimum the top 10 GO terms from IPA with their consensus fold direction would be highly informative. Would also be illustrative and helpful to the reader for the authors to label and highlight individual genes with the largest fold changes that are hyper significant in the figure S1A volcano plot.

We have amended that data accordingly with significant GO terms from the *WebGestalt*-based GO analysis and a volcano plot with highlighted individual genes (Figure 3-Figure suppl. 1). As the IPA-based GO-analysis is not sufficiently convincing, we did not include it.

Line 194-195 The rational for Sept10 is a pretty weak statement, what about other proteins like Tprn or Mkl2 which are more specific and abundant? Even Spef2 (sperm flagellar protein) would have been much more interesting given phenotype observed. Authors should either elaborate further or modify.

Mkl2 was considered as a lower priority candidate as only 3 tryptic peptides could be detected in the Adgb IP and the sequence coverage was very low (4%); moreover, the abundance of Mkl2 was also below the median protein abundance in the Adgb IP. Based on its abundance in the Adgb IP and the relatively low sequence coverage (11.5%), Spef2 was also regarded as a less obvious candidate than Sept10. Considering its established role in spermatogenesis, we have nevertheless tested this potential interaction by co-IP. However, we could not confirm the interaction between Adgb and Spef2, most likely for technical reasons due to poor quality of the commercial antibodies purchased from Σ (catalogue number WH0079925M1) and Bioss (catalogue number bs^-1^1488R). Similarly, the sequence coverage for Tprn (26.7%) was lower compared to the Septs 2, 7, 10, and 11, which consistently all display considerable larger sequence coverage, similar to the range of Adgb (47.9%):

Sept7: 17 peptides and 40.6% sequence coverage

Sept2: 17 peptides and 58.2% sequence coverage

Sept11: 13 peptides and 33.4% sequence coverage

Sept10: 15 peptides and 40.5% sequence coverage

In any case, we will test the potential interactions between Adgb and Spef2 and Tprn, respectively, in follow-up studies.

Lines 214-227 For IFs of spermatids figure 5, in general localization of Adgb from images is difficult to appreciate, appears to be a large amount of unspecific staining. At a minimum, KO spermatids should be used as a negative control for early stage (s7, 12) as they are still present and will help clarify (2nd only panel is not sufficient, will never see any signal and you have not shown those same stages). Additionally, subpanel B does not enhance the figure substantially given that the sperm are already unhappy at this point. Perhaps show CoxIV staining at earlier stages than mature epididymal sperm.

We have repeated the IF analyses on intact sections (Figure 5A) as requested by reviewer 2 and we have included KO samples as negative control in addition to 2^nd^ antibody-only negative controls. Similarly, we included now also KO samples as negative control for spermatid staining (Figure 5B and Figure 5-Figure suppl. 1A). We shifted the former panel B to the Supplemental material and, as suggested by the reviewer, we included CoxIV staining at earlier stages (Figure 5-Figure suppl. 1B).

Line 240 The introduction of calmodulin requires more explanation or background, it is only mentioned in the introduction when discussing Adgb's structure once as a putative motif and then becomes a major focus for the rest of the manuscript and discussion.

We have now elaborated on calmodulin in the context of spermatogenesis in the introduction section.

Of note, Figure 6D is one of the most striking pieces of data in the entire manuscript, very nice! But how does ADGB perform this function? Does this occur through its N-terminal protease domain?

We appreciate the reviewers positive remark. We have accordingly generated a novel deletion construct and now show that the N-terminal ADGB protease domain is crucial for the downstream effect on SEPT10 cleavage. These experiments have now been included in Figure 6D and contribute to additional mechanistic insights into the ADGB-dependent proteolysis of SEPT10.

Lines 244-247 Why does the full length not interact and why is the data not show? Authors should comment on this further, additionally the data shown from the interaction with the truncated globin gfp is not convincing that this is a "robust" interaction and needs to be supported by the expression of V5 calmodulin alone control which is absent from figure 6E.

As suggested by the reviewer we added an experiment demonstrating the lack of interaction between full-length ADGB and CaM (Figure 6-Figure suppl. 2A). While the original IP was performed only with a single ADGB antibody, we now include a FLAG antibody-based precipitation. Calmodulin binding could be detected in neither of the IPs, despite the convincing input controls. For reasons of consistency and as a fully independent approach we also performed mammalian 2-hybrid experiments. To exclude potential bias in VP16- or Gal4-tags we generated reciprocal fusion proteins for this experiment. These additional experiments with VP16-ADGB full-length and Gal4-CaM as well as with Gal4-ADGB full-length and VP16-CaM confirmed the lack of interaction between full-length ADGB and CaM (Figure 6-Figure suppl. 2B).

We agree with the reviewer that more explanation is required. We hypothesize that the lack of interaction upon overexpression of full-length fusion proteins, is due to interference by the larger protein (particularly the larger 700 amino acid long C-terminal region of unidentified origin) and that the globin domain might be liberated possibly by proteolytic cleavage. This hypothesis is consistent with our finding of time-dependent truncation of full-length ADGB from baculovirally infected *Spodoptera frugiperda* (*Sf9*) insect cells (Bracke et al., 2018, *Anal. Biochem* 543:62-70) as well as e.g. upon overexpression of full-length ADGB in HEK293 cells, resulting in several bands of lower molecular weight (e.g. Figure 6A). Furthermore, this observation is consistent with the nature of calpains which display frequent auto-proteolysis for activation of their protease functionality. We have adapted the text accordingly by adding a paragraph in the discussion. We believe that the central message of the manuscript would not benefit from even further experiments on this subject.

We are grateful to the reviewer for the remark on the V5-calmodulin-alone control, which is actually present (as evidenced by a band in the input middle lane) but was unfortunately mislabeled in Figure 6E. We have corrected this error in Figure 6E. Similarly, in Figure 6F and Figure 6-Figure suppl. 4 (as well as in new Figure 6-Figure suppl. 2A, Figure 6-Figure suppl. 5D and Figure 6-Figure suppl. 5E) the correctly labeled V5-calmodulin-alone control is consistently present.

Lines 250-252 Experiments from figures 6F and S10 are nicely done but what happens when you modify heme content? Deplete heme from the media or block heme synthesis? How do you know these mutant truncated globins aren't still hemylated? And what happens when you make these same mutations in the full length Adgb? All these results demonstrate is that these specific constructs still interact in vitro… it is not indicative that heme coordination is expendable.

We appreciate the positive comment of the reviewer. To address this query, we have now performed experiments under heme depletion and heme synthesis blocking conditions. Despite regulated expression levels of several MMP genes (MMP1, MMP9 and MMP13), the ADGB-CaM interaction was maintained, further substantiating the robustness of our results. These additional experiments are integrated in Figure 6-Figure suppl. 5 and we have reported these observations now in the Results section. As mentioned above, full-length ADGB likely interferes with CaM binding. Therefore, we did not attempt to perform these experiments upon mutation of the full-length protein.

Lines 259-263 and figure 6 legend describing the mammalian-2 hybrid assays under normoxic or hypoxic conditions – the authors claim exposure to hypoxia does not alter the luciferase reporters. However, in the legend they state that they do not include the Gal4-CaM alone control condition because of its hypoxic regulation. How then is this a good model system? And how do the authors explain the single transfection alone being modulated by hypoxia but the dual expression not? If anything, it is an essential control, particularly in comparison to the Epo construct they have used as a hypoxic positive control. This data should not be omitted and should be shown and explained/interpreted.

Based on the reviewer’s suggestion we have re-inserted the single Gal4-CaM control in Figure 6H. Several studies reported that low oxygen levels result in an increase of cytoplasmic calcium mobilized from intracellular stores or by extracellular influx, which leads to the activation of calmodulin and its downstream effectors (Arnould et al., 1992, *J. Cell. Physiol.* 152, 215-221; Jung et al., 2010, *J. Biol. Chem.* 285, 25867-25874; Kanatous et al., 2009, *Am. J. Physiol. Cell. Physiol.* 296, C393-402; Soderling and Stull, 2001, *Chem. Rev.* 101, 2341-2352; Stull, 2001, *J. Biol. Chem.* 276, 2311-2312). Moreover, Yuan and colleagues have shown a pronounced and significant response using an HRE-luciferase construct under hypoxic conditions, which was inhibited by increasing doses of the calmodulin inhibitor W-7 (Yuan et al., 2005, *J. Biol. Chem.* 280, 4321-4328). Similarly, in our experiment we observed a significantly increased luciferase activity with overexpressing calmodulin alone under normoxic vs. hypoxic conditions, whereas there was no difference for the globin alone. Furthermore, in the presence of 300 ng of Gal4-CaM and 500 ng of VP16-globin domain expression vectors under hypoxic conditions, we still observed an augmented signal as compared to calmodulin alone, albeit not significant. The luciferase reporter gene-based mammalian-2 hybrid technique is generally seen as a sensitive approach and one of the advantages is its quantifiability. The *EPO* gene is a strong hypoxia-inducible control and the Gal4-CaM-alone control is only moderately induced in comparison to *EPO*. We are convinced that this result does not influence our conclusion. Due to the comment by reviewer 2 on the length of the discussion, and because we do not believe that this is a central message of our paper, we have not elaborated on this effect in the discussion, apart from referring to it explicitly in the Results section.